# FedNolowe: A normalized loss-based weighted aggregation strategy for robust federated learning in heterogeneous environments

**Duy-Dong Le**[1], **Tuong-Nguyen Huynh**[1*], **Anh-Khoa Tran**[2], **Minh-Son Dao**[2], **Pham The Bao**[3]

**1** Industrial University of Ho Chi Minh City (IUH), Ho Chi Minh City, Vietnam, **2** National Institute of Information and Communications Technology (NICT), Koganei, Japan, **3** Saigon University (SGU), Ho Chi Minh City, Vietnam

* htnguyen@iuh.edu.vn

**Data availability statement:** All relevant data are within the manuscript.

## Abstract

Federated Learning supports collaborative model training across distributed clients while keeping sensitive data decentralized. Still, non-independent and identically distributed data pose challenges like unstable convergence and client drift. We propose Federated Normalized Loss-based Weighted Aggregation (FedNolowe) (Code is available at https://github.com/dongld-2020/fednolowe), a new method that weights client contributions using normalized training losses, favoring those with lower losses to improve global model stability. Unlike prior methods tied to dataset sizes or resource-heavy techniques, FedNolowe employs a two-stage L1 normalization, reducing computational complexity by 40% in floating-point operations while matching state-of-the-art performance. A detailed sensitivity analysis shows our two-stage weighting maintains stability in heterogeneous settings by mitigating extreme loss impacts while remaining effective in independent and identically distributed scenarios.

## Introduction

Federated Learning (FL) offers a groundbreaking framework for training machine learning models across a wide array of decentralized devices, introducing an effective strategy to protect data privacy by eliminating the reliance on centralized data storage systems [1]. This innovative method has gained traction in critical fields such as healthcare [2–4], smart agriculture [5], mobile computing [6], and blockchain technology [7], where safeguarding sensitive data and adhering to strict regulatory requirements are of utmost importance. Nevertheless, FL faces significant challenges, largely due to statistical heterogeneity caused by non-independent and identically distributed (non-i.i.d.) data across participating clients [8]. This diversity leads to pressing problems, such as inconsistent convergence, client drift, and the risk of producing biased global models [9,10].

**Funding:** The author(s) received no specific funding for this work.

**Competing interests:** The authors have declared that no competing interests exist.

The seminal FedAvg (Federated Averaging, invented by H. Brendan McMahan et al. (2017) [1].) algorithm [1] aggregates client updates using weights proportional to local dataset sizes. One of the main advantages of FedAvg is its ability to reduce communication rounds by allowing multiple local updates before aggregation. However, FedAvg struggles when dealing with non-i.i.d data distributions among clients, leading to slower convergence and poor generalization [11,12]. This limitation has spurred the development of advanced FL aggregation strategies and algorithms. FedProx (Federated Proximal Optimization, introduced by Tian Li et al. (2018) [13].) [13] introduces a proximal term to constrain local updates, mitigating drift but increasing computational cost. FedMa (Federated Maching Averaging, created by Hongyi Wang et al. (2020) [14].) [14] employs layer-wise neuron matching to align heterogeneous models, achieving robust performance at the expense of scalability. More recent methods, such as FedAsl (Federated Learning with Auto-weighted Aggregation based on Standard Deviation of Training Loss, developed by Zahidur Talukder et al. (2022) [15].) [15], FedLaw (Federated Learning with Learnable Aggregation Weights, devised by Zexi Li et al. (2023) [16].) [16], and A-Flama (Accuracy-based Federated Learning with Adaptive Model Aggregation, innovated by Rebekah Wang et al. (2024) [17].) [17] dynamically adjusting weights based on loss statistics or proxy datasets on the server-side can improve performance. Still, these approaches add complexity or require server-side dependencies, making deployment challenging in resource-limited environments.

Drawing on concepts from correlation-based weighting [18] and FedAsl [15], this paper presents Federated Normalized Loss-based Weighted (FedNolowe)—a streamlined and efficient aggregation technique designed to address data heterogeneity FL. FedNolowe assigns dynamic weights to clients based exclusively on their training losses, employing a two-step L1 normalization process, also known as Least Absolute Deviations (LAD) or Manhattan normalization. In the first step, the training losses are normalized to the open interval (0, 1) using the standard L1 norm. In the second step, these normalized losses are inverted (using 1 minus the normalized value) and then subjected to L1 normalization again. This method enhances the influence of high-performing clients, is less sensitive to outliers, and still maintains computational simplicity. Although FedNolowe does not eliminate the impact of clients with high noise, it significantly reduces their effect on the aggregation, fostering a more equitable and resilient FL system. Experimental results show that FedNolowe surpasses existing state-of-the-art approaches. The primary contributions of this work are as follows:

- We introduce a novel loss-based weighting mechanism that normalizes client losses in two stages, where the second stage applies a subtraction-based L1 normalization, ensuring robustness to non-i.i.d data without additional constraints or server-side resources.
- We validate FedNolowe through extensive experiments on benchmark datasets (MNIST, Fashion-MNIST, and CIFAR-10), demonstrating competitive performance and up to 40% reduction in computational complexity compared to leading methods.
- We conduct a detailed sensitivity analysis comparing our two-stage weighting scheme to alternative approaches. The result shows that it maintains stability in heterogeneous settings by mitigating the impact of extreme loss values while preserving effectiveness in i.i.d scenarios.
- We provide a theoretical convergence analysis under standard FL assumptions, proving that FedNolowe converges to a stationary point of the global loss function.

The paper is structured as follows: Section Related Work reviews related work, Section Proposed FedNolowe details the FedNolowe methodology, Section Experiments Setup describes the experimental setup, Section Results presents results, and Section Conclusion concludes

with future directions. The sensitivity analysis comparing the inversion methods in Fed-Nolowe and FedAsl can be found in Appendix Sensitivity Analysis of FedNolowe and FedAsl, while the detail analysis of FedNolowe's convergence is provided in Appendix Detail Convergence Analysis of FedNolowe.

## Related work

FL has evolved significantly since the introduction of FedAvg by McMahan et al. [1], which aggregates client updates using weights proportional to dataset sizes, i.e., $w^{t+1} = \sum_{k \in S^t} \frac{n_k}{n} w_k^{t+1}$, where $w_k^{t+1}$ is the local model of client $k$ after training on its dataset size $n_k$, $n = \sum_k n_k$, and $S^t$ is a subset of clients that participate in the training round $t$. While effective in reducing communication overhead by allowing multiple local updates, FedAvg struggles with non-i.i.d data, resulting in slow convergence and model bias [10,11,19]. This has prompted a rich body of research aimed at addressing statistical heterogeneity in FL.

One prominent approach is FedProx [13], which mitigates client drift by adding a proximal term to the local objective, formulated as $\min_w(F_k(w) + \frac{\mu}{2}\|w - w^t\|^2)$, where $w^t$ is the global model and $\mu$ controls regularization strength. The server aggregates updates simply as $w^{t+1} = 1/|S^t| \sum_{k \in S^t} w_k^{t+1}$. While FedProx improves stability under non-i.i.d conditions, its additional computation per client increases overhead, particularly for resource-constrained devices [20]. Similarly, Scaffold (Stochastic Controlled Averaging for Federated Learning, conceived by Sai Praneeth Karimireddy et al. (2019)[20].) [20] employs control variates to correct local gradients, achieving faster convergence but requiring persistent state maintenance across rounds, complicating implementation.

FedMa [14] aligns client models via layer-wise neuron matching using the Hungarian algorithm [21], aggregating weights as $w^{t+1} = 1/|S^t| \sum_{k \in S^t} w_k^{t+1} \Pi_k^T$, where $\Pi_k^T$ is the transposed form of the permutation matrix $\Pi_k$, used to rearrange client $k$'s weights to be consistent with the global model. This method excels with heterogeneous architectures but incurs a high computational cost, limiting scalability to large client pools or complex models [22].

Dynamic weighting strategies have also gained traction. FedAsl [15] dynamically assigns weights to client updates using the standard deviation of their training losses, with the global model updated as $w^{t+1} = \sum_{k=1}^N A_k w_k^{t+1}$, where $A_k = d_k^{-1}/\sum_{k=1}^K d_k^{-1}$. The term $d_k$ is set to $d_k = \beta\sigma$ if the client's loss $L_k$ falls within the "good region" $[\mu_L - \alpha\sigma, \mu_L + \alpha\sigma]$, or $d_k = |L_k - \mu_L|$ otherwise. Here, $\mu_L$ denotes the median loss across all clients, $\sigma$ is the standard deviation of the losses, and $0 < \beta \leq \alpha$ are tunable parameters. This method improves resilience against data heterogeneity, though it remains vulnerable to extreme outlier losses, which may distort the aggregation process. Its training parameters need to be tuned to find the appropriate sets for each dataset and model.

Some other adaptive weighting methods, like FedLaw [16] learn two global parameters: a shrinkage factor $\gamma = \exp(g)$ and a weight vector $\lambda = \text{softmax}(x)$, where $x$ is updated via gradient descent on the proxy dataset. The final aggregation is given by $w^{t+1} = \gamma \sum_{i=1}^N \lambda_i w_i$, where $\gamma$ controls global weight decay, and $\lambda_i$ represents learned client importance scores. FedA-Flama [17] uses accuracy-based weighting, where clients with higher accuracy on the server's test data are assigned higher weights. However, this method requires all clients to be validated with the server's full test dataset, increasing resource usage. Additionally, choosing an appropriate minimum aggregation weight (minAW) threshold and replacing client weights minAW if they exceed the threshold can introduce unfairness issues in FL. While adaptive, it relies on validation clients using server-side data, which increases computational overhead. [23] presents a dynamic node matching FL that outperforms FedAvg, but its complexity may limit scalability with many nodes or clients. Adaptive optimization methods, such as those in [24],

refine aggregation by tuning hyperparameters like learning rates across clients, though they increase communication and computational burdens.

Recent efforts explore alternative paradigms such as. Personalized FL approaches [25,26], decouple global and local objectives to tailor models to individual clients, sacrificing global generalization for personalization. Knowledge distillation-based methods [18,27,28] transfer learned representations across clients, but require a pre-trained teacher model using curated public datasets. Another paper using pre-trained models is [29], which applies a Genetic Algorithm (GA) to optimize FL for sports image classification. The GA improves inference time and reduces storage, but requires more hyperparameter tuning.

This research aims to implement a mechanism that influences model aggregation in FL without requiring additional data or validating client models. Accordingly, we will compare FedNolowe with existing methods, as presented in Table 1.

Table 1 shows the mathematical formulations of the server and client sides, along with the pros and cons, and the applications of state-of-the-art FL algorithms. Our proposed FedNolowe distinguishes itself by relying solely on training losses. It aims to deal with highly non-i.i.d that FedAvg fails to, avoiding the complexity of proximal terms in FedProx and neuron matching in FedMa. Unlike FedAsl, it employs a simpler two-stage normalization that mitigates outlier effects without statistical thresholds. Compared to other methods, it imposes no assumptions on optimization parameters, enhancing flexibility. Our approach thus bridges the gap between performance and efficiency, offering a scalable solution for heterogeneous FL, as validated theoretically and empirically in subsequent sections and appendices.

## Proposed FedNolowe

## Problem formulation

We consider an FL system with $N$ clients, indexed by $k \in \{1, 2, \dots, N\}$, each possessing a local dataset $\mathcal{D}_k$ with $n_k = |\mathcal{D}_k|$ samples drawn from a non-i.i.d distribution $\mathcal{P}_k$. The total dataset size is $n = \sum_{k=1}^{N} n_k$. Each client $k$ trains a local model parameterized by $w \in \mathbb{R}^d$ to minimize its local loss function $\mathcal{L}_k(w) = \frac{1}{n_k} \sum_{x \in \mathcal{D}_k} \ell(w; x)$, where $\ell(w; x)$ is the loss for a single data

**Table 1. Comparison of federated learning algorithms.**

| Method | Server-side (aggregate $w^{t+1}$) | Client-side (update $w_k^{t+1}$) | Characteristics | Applications |
|---|---|---|---|---|
| **FedAvg** [1] | $\sum_{k \in S^t} \frac{n_k}{n} w_k^{t+1}$ | $w_k^t - \eta \nabla \mathcal{L}_k(w_k^t)$ | Simple, communication-efficient; Poor performance on non-i.i.d data | Mobile keyboards [6], healthcare [30], IoT [31] |
| **FedProx** [13] | $\frac{1}{|S^t|} \sum_{k \in S^t} w_k^{t+1}$ | $w_k^t - \eta(\nabla \mathcal{L}_k(w_k^t) + \mu(w_k^t - w^t))$ | Robust to non-i.i.d, reduces drift; Increased client computation | Healthcare [32], finance [33] |
| **FedMa** [14] | $\frac{1}{|S^t|} \sum_{k \in S^t} w_k^{t+1} \Pi_k^T$ | $w_k^t - \eta \nabla \mathcal{L}_k(w_k^t)$ | Handles architecture heterogeneity; High computational cost | Image classification [14], AGNews dataset [34] |
| **FedAsl** [15] | $\sum_{k=1}^{N} A_k w_k^{t+1}$ | $w_k^t - \eta \nabla \mathcal{L}_k(w_k^t)$ | Dynamic weighting by standard deviation of clients training losses; Sensitive to loss outliers. | Tested with MNIST, CIFAR-10, FEMNIST |
| **FedNolowe** | $\sum_{k=1}^{N} \alpha_k w_k^{t+1}$ | $w_k^t - \eta \nabla \mathcal{L}_k(w_k^t)$ | Dynamic loss-based weighting with 2 stages L1 normalization; Can not mitigate all high loss clients | Tested with MNIST, CIFAR-10, Fashion-MNIST |

sample $x$. The global objective is to optimize a weighted combination of local losses:

$$\min_{w \in \mathbb{R}^d} \mathcal{L}(w) = \sum_{k=1}^{N} \alpha_k \mathcal{L}_k(w), \tag{1}$$

where $\alpha_k > 0$ and $\sum_{k=1}^{N} \alpha_k = 1$ represent the contribution weights of client $k$. In FedAvg [1], $\alpha_k = \frac{n_k}{n}$, which struggles under non-i.i.d conditions as local optima diverge from the global optimum [10,35]. In FedAsl [15], $\alpha_k = A_k = \frac{1/d_k}{\sum_{j=1}^{N} 1/d_k}$, which is sensitive to outlier losses. We discuss this limitation in Appendix .

In practice, FL operates in communication rounds. At round $t$, a subset $S^t \subseteq \{1, 2, \ldots, N\}$ of clients are randomly selected, each performing local optimization on the current global model $w^t$. The server aggregates these updates to form $w^{t+1}$. Non-i.i.d data exacerbates client drift, where local updates $w_k^{t+1}$ deviate significantly from $w^t$, degrading performance [9]. FedNolowe addresses this by dynamically adjusting $\alpha_k^t$ based on training loss, prioritizing clients that align better with the global objective.

## FedNolowe weighting mechanism

To tackle heterogeneity, we propose FedNolowe, a two-stage weighting mechanism that leverages local training losses $\mathcal{L}_k^t$ at round $t$. Unlike FedAvg's static weights or FedProx's proximal constraints [13], FedNolowe uses a loss-based approach inspired by correlation weighting in [18], avoiding complex statistical measures (e.g., FedAsl [15]) or server-side proxies (e.g., FedLaw [16], FedA-Flama [17]). Compared to FedAsl's division-based inversion $A_k = 1/\mathcal{L}_k^t$, FedNolowe introduces a fundamentally different two-stage normalization strategy. It first normalizes losses across clients to obtain $\tilde{\mathcal{L}}_k^t$, then applies a subtraction-based inversion $1 - \tilde{\mathcal{L}}_k^t$. This avoids instability caused by division when losses are close to zero or highly skewed—conditions common in non-i.i.d. settings—resulting in more stable and bounded weights.

**Definition 1** (**Computation and Normalized Loss Weights**). *For each client $k \in S^t$ at round $t$:*

1. ***Loss computation:*** *Each local training loss is computed as:*

$$\mathcal{L}_k^t = \frac{1}{E} \sum_{e=1}^{E} \frac{1}{|\mathcal{B}_e|} \sum_{x \in \mathcal{B}_e} \mathcal{L}_k(w_k^t; x) \tag{2}$$

*where $\mathcal{B}_e \subset \mathcal{D}_k$ is a mini-batch of data from the local dataset $\mathcal{D}_k$ of client $k$, and $e$ is a single epoch of the total $E$ epochs.*

2. ***Two Stages Loss Normalization:***

$$\alpha_k^t = \frac{1 - \tilde{\mathcal{L}}_k^t}{\sum_{j \in S^t} \left(1 - \tilde{\mathcal{L}}_j^t\right)} \quad \text{where} \quad \tilde{\mathcal{L}}_k^t = \frac{\mathcal{L}_k^t}{\sum_{j \in S^t} \mathcal{L}_j^t}. \tag{3}$$

*In Eq (3), the term on the right is the first-stage normalization, ensuring scale invariance across clients with varying loss magnitudes. The term on the left $1 - \tilde{\mathcal{L}}_k^t$ inverts the normalized loss, amplifying the influence of clients with lower losses while second-stage normalizing the weights to sum to 1.*

The resulting global update is:

$$w^{t+1} = \sum_{k \in S^t} \alpha_k^t w_k^{t+1}. \tag{4}$$

This mechanism ensures that clients with lower $\mathcal{L}_k^t$, indicative of better local optimization or less drift, contribute more to $w^{t+1}$, enhancing stability without additional computational burdens like neuron matching [14] or variance tracking [20]. Algorithm 1 outlines the full procedure.

**Algorithm 1 FedNolowe: Normalized loss-based weighted aggregation.**

```
 1: ServerSide:
 2: Initialize global model w⁰
 3: for each round t = 0, 1, ..., T − 1 do
 4:     Randomly select subset Sᵗ ⊆ {1, 2, ..., N}
 5:     for each client k ∈ Sᵗ in parallel do
 6:         (wₖᵗ⁺¹, 𝓛ₖᵗ) ← ClientSide(k, wᵗ)
 7:     end for
 8:     Compute αₖᵗ for each k ∈ Sᵗ using Eq (3)
 9:     Update wᵗ⁺¹ ← ∑ₖ∈Sᵗ αₖᵗ wₖᵗ⁺¹
10: end for

11: function ClientSide(k, wᵗ)
12:     for each local epoch e = 1, ..., E do
13:         Update wₖᵗ⁺¹ ← wₖᵗ − η∇𝓛ₖ(wₖᵗ; 𝓑)
14:     end for
15:     Compute average training loss 𝓛ₖᵗ by Eq (2)
16:     return (wₖᵗ⁺¹, 𝓛ₖᵗ)
17: end function
```

In Algorithm 1, each training round begins with the server randomly selecting a subset of clients. These clients train their local models over multiple epochs on their respective datasets, returning updated models and corresponding loss values. FedNolowe's core strength is its elegant simplicity and flexibility: it dynamically emphasizes clients with superior local convergence by leveraging their training losses $\mathcal{L}_k^t$, using only lightweight scalar operations to compute weights (Eq (3). This approach differs markedly from FedAvg's static aggregation, FedProx's per-client proximal regularization, FedMa's computationally intensive neuron matching across $L$ layers with $d$ parameters, and FedAsl's statistical overhead. FedNolowe incurs a server-side complexity of $O(|S^t|d)$ for the weighted aggregation, augmented by a negligible $O(|S^t|)$ normalization cost, rendering it less resource-demanding than FedProx and FedMa while remaining comparable to FedAsl and FedAvg. We discuss this further in Subsection.

## Convergence analysis

We analyze FedNolowe's convergence under standard FL assumptions [13,24,35] and the methods in [36]. FedNolowe's dynamic weighting mitigates non-i.i.d effects by prioritizing clients with lower losses, ensuring gradient alignment with the global objective.

**Assumption 1 (L-smoothness).** *Each local loss $\mathcal{L}_k(w)$ is L-smooth, i.e., $\|\nabla \mathcal{L}_k(w) - \nabla \mathcal{L}_k(w')\| \le L\|w - w'\|$ for some L>0, for all $w, w' \in \mathbb{R}^d$.*

**Assumption 2 (Bounded Gradient).** *The global loss gradient is bounded: $\mathbb{E}[\|\nabla \mathcal{L}(w)\|^2] \le G^2$, where $G > 0$.*

**Assumption 3** (**Finite Variance**). *Local stochastic gradients $g_k^t$ have bounded variance:* $\mathbb{E}\left[\|g_k^t - \nabla\mathcal{L}_k(w^t)\|^2\right] \leq \sigma^2$.

**Assumption 4** (**Alignment of Weights**). *There exists a constant $\beta > 0$ such that*

$$\mathbb{E}\left[\left\langle \nabla\mathcal{L}(w^t), \sum_{k \in S^t} \alpha_k^t \nabla\mathcal{L}_k(w^t) \right\rangle\right] \geq \beta\|\nabla\mathcal{L}(w^t)\|^2,$$

*where the weights $\alpha_k^t$ are as described in Definition 1.*

**Theorem 1.** *Under Assumptions 1–4, FedNolowe converges to a stationary point, i.e.,* $\lim_{T \to \infty}\mathbb{E}[\|\nabla\mathcal{L}(w^t)\|^2] \to 0$.

We leave the detailed convergence proof in Appendix.

## Experiments setup

To evaluate FedNolowe's effectiveness, we conduct experiments on three benchmark datasets under non-i.i.d settings, comparing its performance and efficiency against the state-of-the-art FL methods. This section details the datasets, data partitioning, model architectures, training parameters, and evaluation metrics, ensuring reproducibility and robustness of results.

### Datasets and non-i.i.d partitioning

We utilize three widely adopted datasets: MNIST [37], Fashion-MNIST [38], and CIFAR-10 [39], each consisting of 10 classes. MNIST contains 60,000 training and 10,000 test grayscale images of handwritten digits (28×28 pixels). Fashion-MNIST mirrors the structure of MNIST but features images of clothing items, making it a more challenging classification task. CIFAR-10 includes 50,000 training and 10,000 test RGB images (32×32 pixels) of various objects (e.g., airplanes, cars), with increased complexity due to color and semantic variations. We split the training data of these three datasets across the clients, as described in Figs 1 and 2, while keeping the test data on the server-side to examine the performance of experimental methods [14].

To simulate non-i.i.d data distributions, we partitioned each dataset across 50 clients using a Dirichlet distribution, a widely adopted method in federated learning research [19]. The concentration parameter $\alpha \in (0, \infty)$ governs the degree of data heterogeneity, with $\alpha \to \infty$ indicating an i.i.d data partition. While FedMA [14] and FedProx [13] employed $\alpha = 0.5$, we set $\alpha = 0.1$ for MNIST to create highly heterogeneous distributions, and $\alpha = 0.2$ for Fashion-MNIST and CIFAR-10 to reflect moderate heterogeneity, accounting for their greater complexity compared to MNIST. This setup results in clients receiving varying sample sizes and uneven class distributions, as depicted in Figs 1 and 2. These Figs illustrate the skewed, non-uniform data allocation typical of real-world federated learning scenarios.

For the class distributions in Fig 1, MNIST(left) exhibits significant heterogeneity with $\alpha = 0.1$, where most clients have uneven class proportions: Clients 10 and 49 show the highest diversity, containing 9 classes, while Clients 1, 2, and 38 have the least diversity, dominated by only one class (3-red, 4-purple, 0-blue), respectively. For Fashion-MNIST (middle, $\alpha = 0.2$), Clients 6, 22, 3, 41, 47, and 49 are the most diverse, featuring 9 classes, whereas Client 29 is the least diverse, only consisting of Class 0-blue with negligible contributions from others. In CIFAR-10 (right, $\alpha = 0.2$), Clients 3 and 49 display the greatest diversity across all 10 classes, while Client 8 is the least diverse, dominated by Class 1-yellow. These patterns underscore the non-i.i.d nature of the data, with varying class concentrations across clients.

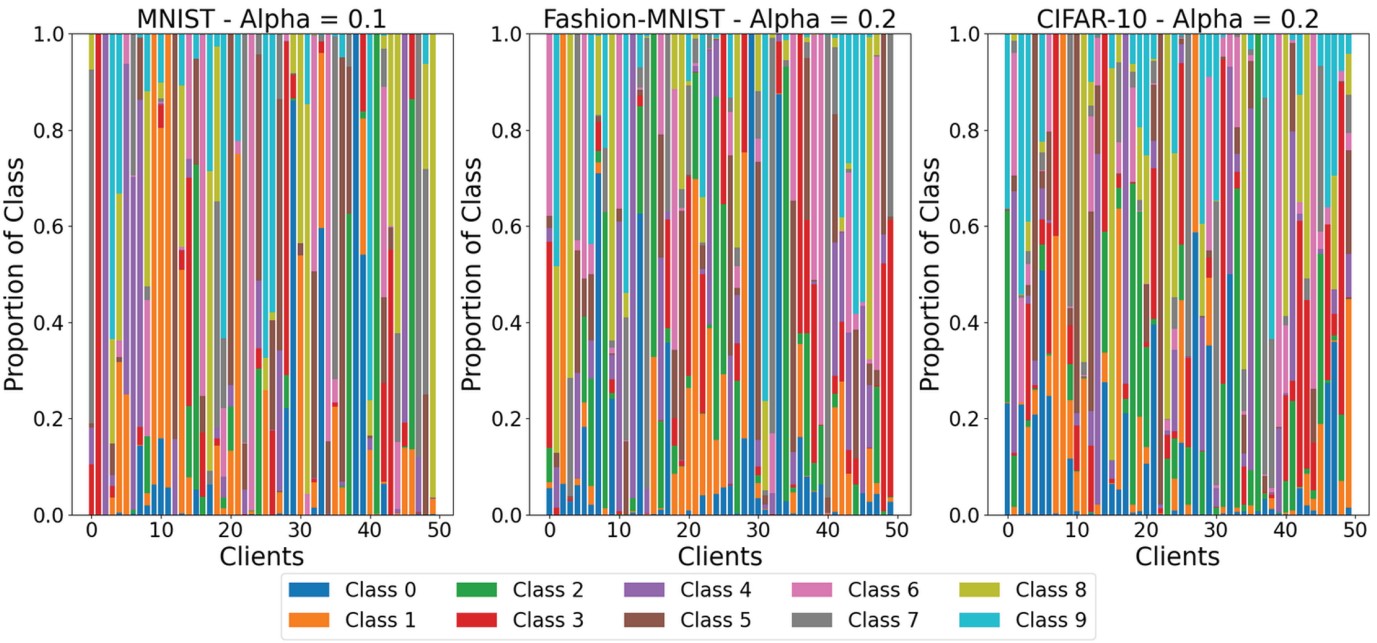

**Fig 1. Class distributions across datasets.**

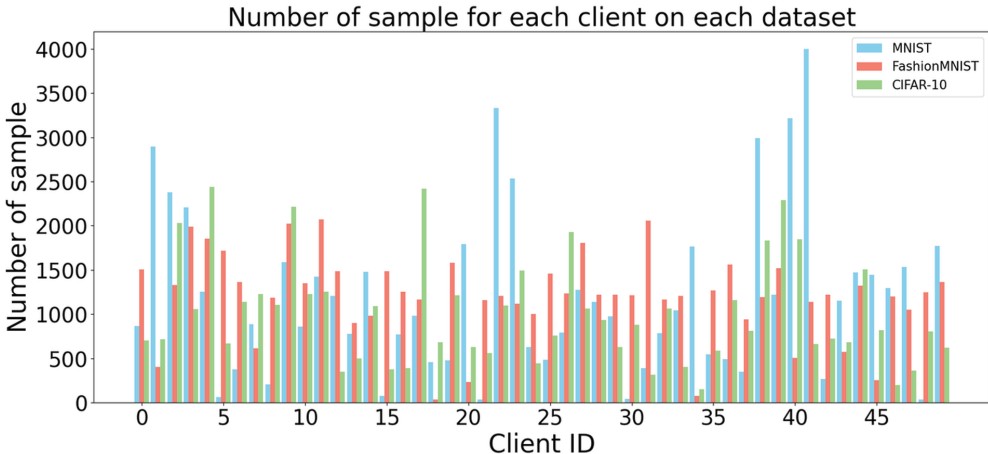

**Fig 2. Sample size distributions across datasets.**

For the sample distributions in Fig 2, the allocation is highly uneven across all datasets due to the Dirichlet distribution. In MNIST, Client 41 has the highest (3997 samples), and Client 48 is the lowest (32 samples). For Fashion-MNIST, Client 11 is peaking at 2070 samples, and Client 18 is the minimum (35 samples). In CIFAR-10, Client 4 has the highest (2436 samples), and Client 34 is the lowest (149 samples). This variability reflects the non-uniform sample distribution, characteristic of real-world heterogeneous FL environments.

Data preprocessing follows standard protocols. MNIST and Fashion-MNIST images are normalized to [0,1] with means 0.1307 and 0.2860, and standard deviations 0.3081 and 0.3530, respectively. Fashion-MNIST training data is augmented with random horizontal

flips (probability 0.5). CIFAR-10 images are augmented with random crops (padding 4), horizontal flips, and color jitter (brightness, contrast, saturation = 0.2), then normalized using per-channel means (0.4914, 0.4822, 0.4465) and standard deviations (0.2023, 0.1994, 0.2010).

## Model architectures

We employ three convolutional neural networks tailored to the complexity of the datasets: LeNet-5 for MNIST, a custom CNN for Fashion-MNIST, and VGG-9 for CIFAR-10. These architectures incorporate batch normalization (BN) and dropout to enhance generalization and mitigate overfitting, balancing computational efficiency and representational capacity under FL's resource constraints.

- **LeNet-5** [40]: Designed for MNIST, this model comprises two convolutional layers and three fully connected layers. The first convolutional layer accepts 1 input channel (grayscale) and produces 32 output channels using a 5×5 kernel, followed by BN, ReLU activation, and 2×2 max-pooling, reducing the spatial dimensions from 28×28 to 14×14. The second convolutional layer takes 32 input channels, yields 64 output channels with a 5×5 kernel, and applies BN, ReLU, and max-pooling, resulting in 64 feature maps of size 4×4. These are flattened into a 1,024-dimensional vector ($64 \times 4 \times 4$) for the fully connected layers. The first fully connected layer (FC1) maps 1,024 inputs to 256 units with ReLU activation and a dropout layer (probability 0.5). The second fully connected layer (FC2) reduces this to 128 units with ReLU activation, followed by the output layer (FC3) mapping to 10 units for the 10-digit classes, without an activation function before softmax computation in the loss function.
- **Custom CNN**: Developed for Fashion-MNIST, this network features three convolutional layers and two fully connected layers. The first convolutional layer processes 1 input channel into 32 output channels with a 3×3 kernel and padding of 1, followed by BN, ReLU, and 2×2 max-pooling, reducing the spatial size from 28×28 to 14×14. The second convolutional layer maps 32 channels to 64 with a 3×3 kernel, BN, ReLU, and max-pooling, yielding 7×7 feature maps. The third convolutional layer increases to 128 channels with a 3×3 kernel, BN, ReLU, and max-pooling, producing 128 feature maps of size 3×3. These are flattened into a 1,152-dimensional vector ($128 \times 3 \times 3$), feeding into a fully connected layer (FC1) of 512 units with ReLU and dropout (probability 0.3), followed by an output layer (FC2) of 10 units without additional activation.
- **VGG-9** [41]: Applied to CIFAR-10, this model consists of three blocks of convolutional layers followed by three fully connected layers. Each block contains two convolutional layers with 3×3 kernels and padding of 1, increasing channels from 3 to 64 (Block 1), 64 to 128 (Block 2), and 128 to 256 (Block 3), with BN and ReLU after each convolution. Block 1 ends with 2×2 max-pooling (32×32 to 16×16), Block 2 reduces to 8×8, and Block 3 to 4×4, yielding 256 feature maps of size 4×4. These are flattened into a 4,096-dimensional vector ($256 \times 4 \times 4$), processed by a fully connected layer (FC1) of 512 units with ReLU and dropout (probability 0.5), a second fully connected layer (FC2) of 512 units with ReLU and dropout (0.5), and an output layer (FC3) of 10 units.

## Training parameters and evaluation metrics

Each experiment simulates an FL system with 50 clients over $T$ communication rounds, randomly selecting a fraction $C \in \{10\%, 20\%, 30\%\}$ (MNIST, Fashion-MNIST) or $\{20\%, 30\%, 40\%\}$ (CIFAR-10) of clients per round. Local training uses stochastic gradient descent (SGD) with learning rate $\eta = 0.01$, momentum 0.9, weight decay 0.001, and batch size

32. Local epochs $E$ are set to 2 (MNIST), 3 (Fashion-MNIST), and 5 (CIFAR-10), reflecting the higher number of epochs for more challenging datasets and the complexity of the model's architecture. Communication rounds are $T = 50$ for MNIST and $T = 100$ for Fashion-MNIST and CIFAR-10. Table 2 summarizes key parameters.

We benchmark FedNolowe against five state-of-the-art FL baselines: FedAvg, which aggregates updates weighted by local dataset sizes; FedProx, incorporating a proximal term with $\mu = 0.001$ to mitigate drift; FedMa, utilizing layer-wise neuron matching via the Hungarian algorithm for heterogeneous models; FedAsl, employing loss deviation weights with parameters $\alpha = 1$ and $\beta = 0.2$ which gave the best experiment results as presented in Fig 9(b) of [15]. Performance is evaluated using four metrics computed on the global test set after each communication round: training loss, averaging the local loss across selected clients; validation loss, assessing the global model on the test set; accuracy, measuring Top-1 classification accuracy; and F1-score, the harmonic mean of precision and recall, accounting for class imbalance in non-i.i.d settings. To obtain a comprehensive perspective across all training rounds, we compute the average of these metrics over all rounds rather than relying solely on the final round's values.

Computational efficiency is measured as floating-point operations (FLOPs) per round using PyTorch's profiler [42], [43]. Local FLOPs are derived from forward and backward passes over $E$ epochs on each client, while aggregation FLOPs account for method-specific operations: weighted averaging (FedAvg, FedNolowe), proximal term computation (FedProx), neuron matching (FedMa), loss statistics (FedAsl).

## Results

We assess the performance of FedNolowe in comparison to four baseline approaches FedAvg [1], FedProx [13], FedMa [14], and FedAsl [15] across the MNIST, Fashion-MNIST, and CIFAR-10 datasets under non-i.i.d conditions. Results are presented as averages derived from three independent runs, emphasizing training and validation loss, accuracy, F1-score, and computational efficiency measured in FLOPs. In the following subsections, we provide graphical comparisons of the progression of training loss (subfigure a), validation loss (subfigure b), and accuracy (subfigure c) for each run with the percentage of randomly selected client subsets, followed by the table computation of average metrics, including F1-score, across all training rounds. Visualizations through Figs and tables highlight performance trends and mean values, with FedNolowe consistently achieving a strong balance between effectiveness and efficiency.

**Table 2. Training parameters across datasets.**

| Parameter | MNIST | Fashion-MNIST | CIFAR-10 |
| --- | --- | --- | --- |
| Model | LeNet-5 | Custom CNN | VGG-9 |
| Clients | 50 | 50 | 50 |
| Dirichlet $\alpha$ | 0.1 | 0.2 | 0.2 |
| Client Fraction ($C$) | 10%, 20%, 30% | 10%, 20%, 30% | 20%, 30%, 40% |
| Local Epochs ($E$) | 2 | 3 | 5 |
| Rounds ($T$) | 50 | 100 | 100 |
| Learning Rate ($\eta$) | 0.01 | 0.01 | 0.01 |
| Batch Size | 32 | 32 | 32 |

## Experiment 1: MNIST

Fig 3 plots the results of over 50 rounds for client fractions $C = 10\%$ on the MNIST dataset. As shown, for all three metrics—training loss, validation loss, and accuracy—FedAvg (yellow) and FedAsl (purple) experience significant fluctuations, while FedNolowe (blue), Fed-Prox (green), and FedMa (red) exhibit similar stable progress. The specific averages of training values will be provided in Table 3, where a more detailed analysis will be presented.

Fig 4 shows the results of over 50 rounds for client fractions $C = 20\%$ on the MNIST dataset. FedAsl (purple) demonstrates fewer fluctuations than the previous result in Fig 3. FedAvg (yellow) still exhibits significant training loss, validation loss, and accuracy oscillations. In contrast, FedNolowe (blue), FedProx (green), and FedMa (red) show stable progress with similar performance trends, indicating better convergence. The specific averages of training values will be provided in Table 3, where a more detailed analysis will be presented.

Fig 5 presents the results of over 50 rounds for client fractions $C = 30\%$ on the MNIST dataset. The performance trends are similar to those observed in the $C = 20\%$ scenario (Fig 4).

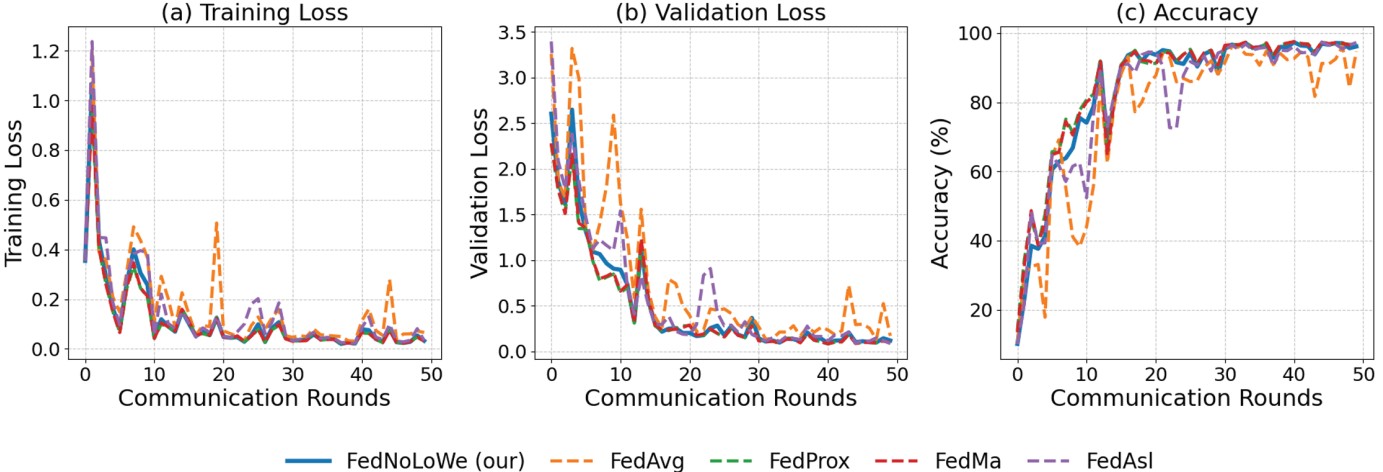

**Fig 3. Performance on MNIST with** $C = 10\%$ **clients per round: (a) training loss, (b) validation loss, (c) accuracy.**

**Table 3. Mean metrics on MNIST across client fractions (Std with rolling window = 5).**

**Train Loss and Validation Loss**

| Algorithm | Train Loss | | | Validation Loss | | |
|---|---|---|---|---|---|---|
| | 10% | 20% | 30% | 10% | 20% | 30% |
| FedProx | 0.11 ± 0.05 | 0.09 ± 0.03 | 0.08 ± 0.02 | 0.46 ± 0.13 | 0.31 ± 0.08 | 0.26 ± 0.07 |
| FedMa | 0.11 ± 0.05 | 0.09 ± 0.03 | 0.08 ± 0.02 | 0.46 ± 0.13 | 0.31 ± 0.08 | 0.26 ± 0.06 |
| FedNolowe | 0.12 ± 0.06 | 0.09 ± 0.03 | 0.08 ± 0.02 | 0.51 ± 0.14 | 0.32 ± 0.08 | 0.26 ± 0.07 |
| FedAsl | 0.15 ± 0.07 | 0.09 ± 0.03 | 0.08 ± 0.02 | 0.60 ± 0.18 | 0.32 ± 0.09 | 0.27 ± 0.07 |
| FedAvg | 0.17 ± 0.09 | 0.12 ± 0.04 | 0.10 ± 0.03 | 0.77 ± 0.30 | 0.56 ± 0.19 | 0.37 ± 0.09 |

**F1-score and Accuracy**

| Algorithm | F1-score | | | Accurary (%) | | |
|---|---|---|---|---|---|---|
| | 10% | 20% | 30% | 10% | 20% | 30% |
| FedProx | 0.84 ± 0.04 | 0.90 ± 0.03 | 0.91 ± 0.03 | 85.29 ± 5.53 | 90.44 ± 2.64 | 91.65 ± 2.36 |
| FedMa | 0.84 ± 0.04 | 0.90 ± 0.03 | 0.91 ± 0.03 | 85.12 ± 4.18 | 90.35 ± 2.68 | 91.65 ± 2.36 |
| FedNolowe | 0.82 ± 0.05 | 0.89 ± 0.03 | 0.91 ± 0.03 | 83.76 ± 4.22 | 90.15 ± 2.77 | 91.54 ± 2.45 |
| FedAsl | 0.80 ± 0.06 | 0.89 ± 0.03 | 0.91 ± 0.03 | 81.69 ± 5.53 | 89.79 ± 2.93 | 91.57 ± 2.52 |
| FedAvg | 0.76 ± 0.08 | 0.81 ± 0.06 | 0.86 ± 0.04 | 77.60 ± 7.32 | 82.09 ± 5.37 | 87.62 ± 3.73 |

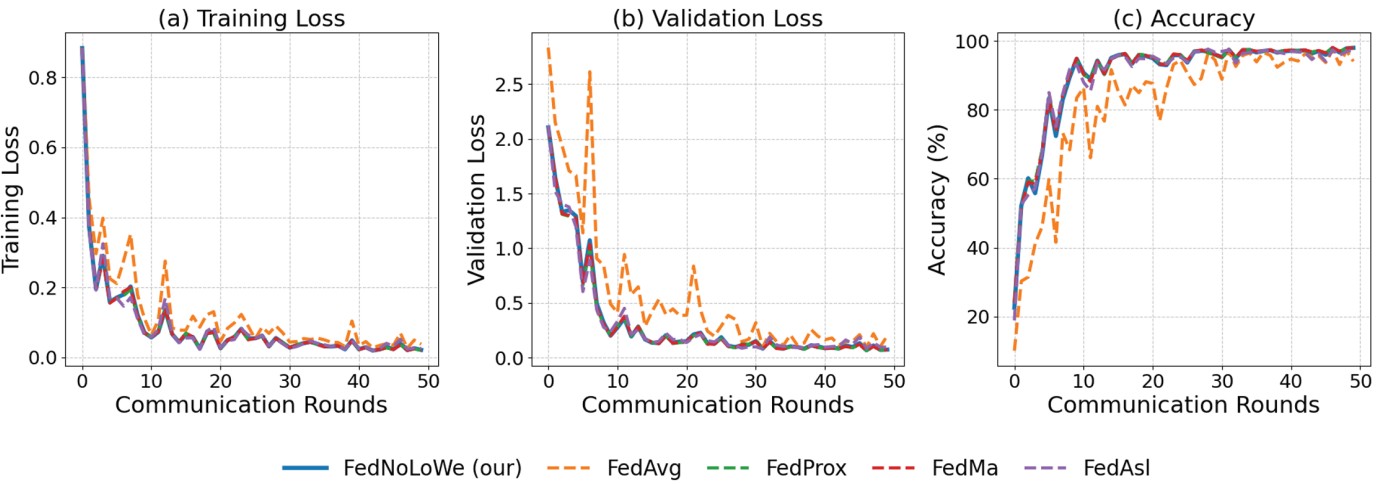

**Fig 4. Performance on MNIST with** $C = 20\%$ **clients per round: (a) training loss, (b) validation loss, (c) accuracy.**

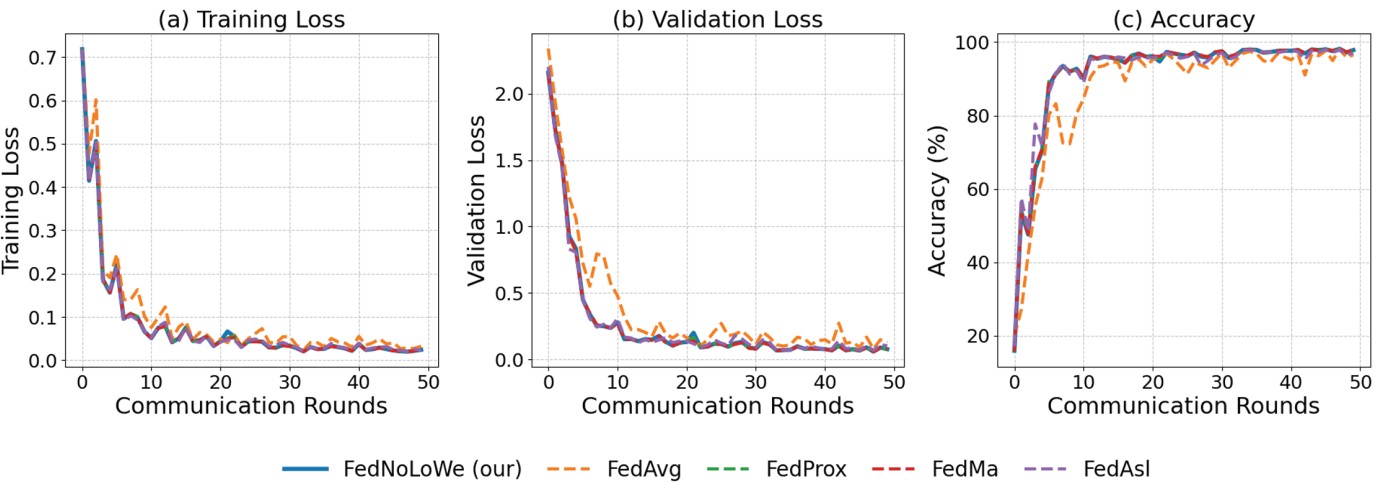

**Fig 5. Performance on MNIST with** $C = 30\%$ **clients per round: (a) training loss, (b) validation loss, (c) accuracy.**

FedAvg (yellow) still exhibits some fluctuation in training loss, validation loss, and accuracy, while the other algorithms — FedNolowe (blue), FedProx (green), and FedMa (red) — show more stable progress. However, with the increased client fraction, all algorithms have become more stable compared to the $C = 10\%$ and $C = 20\%$ scenarios (Fig 3, Fig 4 resp.), indicating that a higher client pool helps in reducing fluctuations and improving convergence. These results further emphasize the benefit of increasing the number of clients to achieve more stable and reliable FL performance. The specific averages of training values will be provided in Table 3, where a more detailed analysis will be presented.

Table 3 shows the mean performance of FL algorithms on the MNIST dataset across different client fractions ($C = 10\%$, 20%, and 30%). FedProx and FedMa consistently achieve the lowest training and validation losses, with FedProx slightly leading in accuracy (91.65±2.36%) and F1-score (0.91 ± 0.03) at $C = 30\%$. FedNolowe performs comparably, with slightly higher losses and accuracy of 91.54 ± 2.45%. In contrast, FedAsl and FedAvg lag behind. FedAsl

exhibits higher losses and marginally lower accuracy (91.57 ± 2.52%), while FedAvg shows the greatest instability, with the highest training loss (0.17 ± 0.09) and lowest accuracy (77.60 ± 7.32%) at $C$ = 10%.

With these results, it can be concluded that on the MNIST dataset (non-i.i.d, $\alpha$ = 0.1), Fed-Prox and FedMa deliver the best performance, with FedNolowe following closely behind. FedNolowe shows competitive results, especially in accuracy and F1-score, although slightly trailing FedProx and FedMa. In contrast, FedAsl and FedAvg are less effective, particularly at lower client fractions, with FedAvg showing significant instability.

### Experiment 2: Fashion-MNIST

Figs 6, 7, and 8 show the performance of FL algorithms on the Fashion-MNIST dataset with client fraction ($C$ = 10, 20, 30% respectively) across 100 communication rounds. The performance of the five algorithms follows a similar pattern to the results on MNIST. FedNolowe, FedProx, and FedMa maintain comparable performance, showing stable training and validation losses, as well as high accuracy. On the other hand, FedAvg and FedAsl exhibit more fluctuations and less stability, particularly in the early rounds. However, the performance gap between the two groups is not as large as observed on MNIST, indicating that while FedAvg and FedAsl are still less stable, their relative disadvantage is less pronounced on Fashion-MNIST. Table 4 reported more details.

Table 4 shows the performance of FL algorithms on the Fashion-MNIST dataset with client fractions of 10%, 20%, and 30%. FedProx and FedMa consistently outperform the other algorithms, achieving the lowest training and validation losses (0.10 ± 0.01 and 0.38 ± 0.05 at 30%) and the highest F1-score (0.86 ± 0.02) and accuracy (86.63 ± 1.53%). FedNolowe follows closely behind, with slightly higher losses and F1-score (0.86 ± 0.02) and accuracy (86.50 ± 1.57%). FedAvg and FedAsl show weaker performance, with FedAsl particularly struggling with higher losses and lower accuracy (84.54 ± 2.04%), highlighting its instability compared to the others.

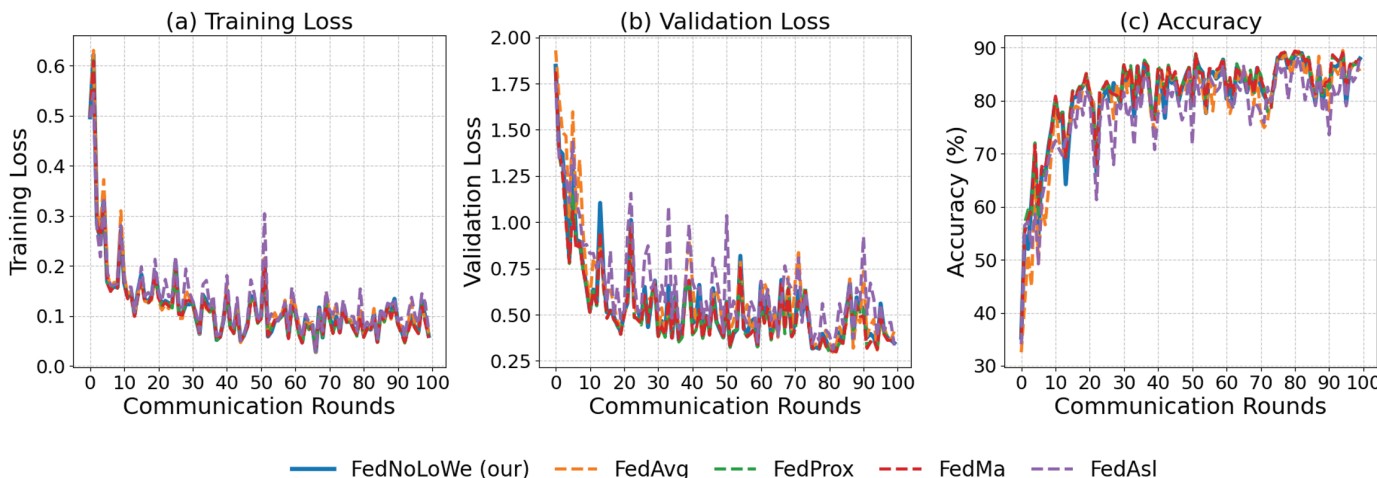

**Fig 6. Performance on Fashion-MNIST with** $C$ = 10% **clients per round: (a) training loss, (b) validation loss, (c) accuracy.**

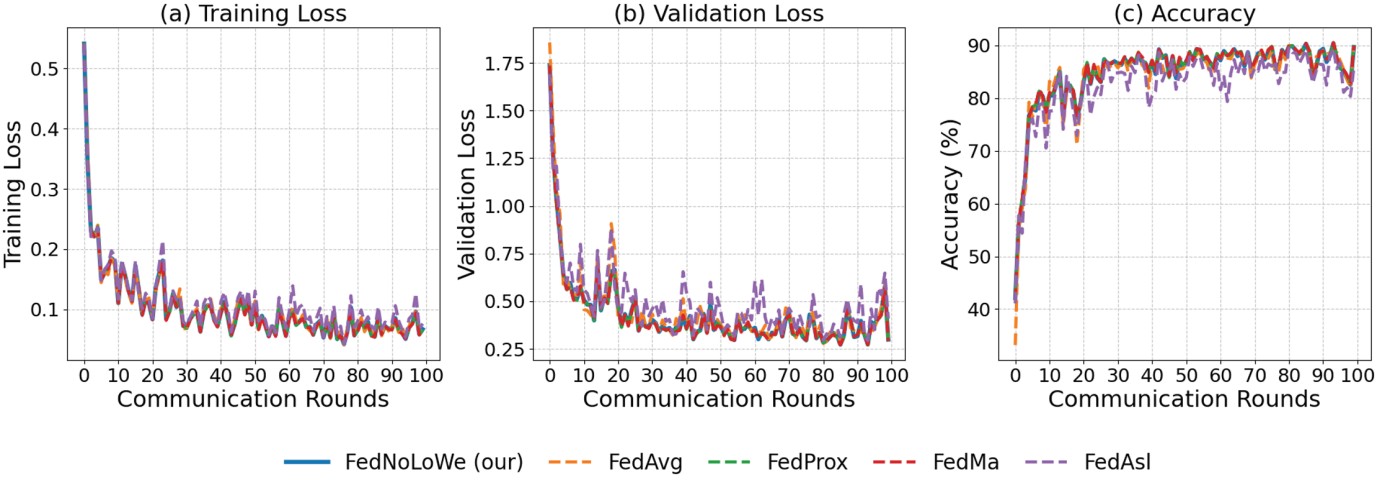

**Fig 7. Performance on Fashion-MNIST with** $C$ = 20% **clients per round: (a) training loss, (b) validation loss, (c) accuracy.**

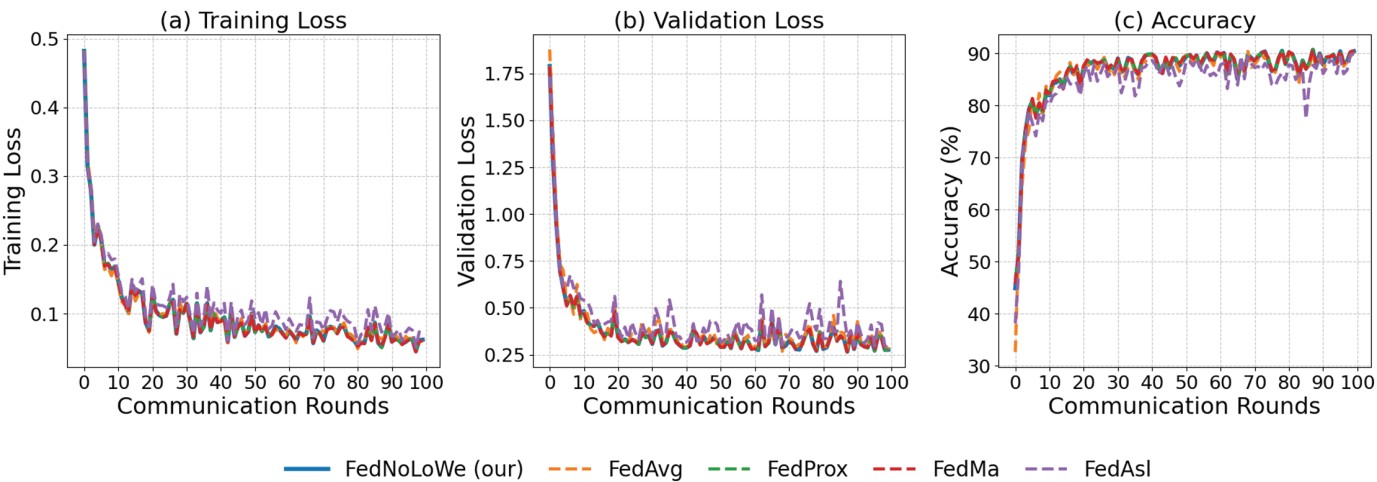

**Fig 8. Performance on Fashion-MNIST with** $C$ = 30% **clients per round: (a) training loss, (b) validation loss, (c) accuracy.**

### Experiment 3: CIFAR-10

Figs 9, 10, and 11 and Table 5 show results over 100 rounds with $C$ = 20%, 30%, and 40%. Fed-Nolowe and FedProx lead, with FedNolowe achieving the lowest validation loss ($0.92 \pm 0.05$ at $C$ = 40%). FedNolowe's accuracy ($67.61 \pm 2.17$%) matches FedProx and is 8.46% higher than FedMa's ($1.14 \pm 0.05$ loss, $59.15 \pm 2.17$% accuracy). FedAvg and FedAsl exhibit moderate performance, with FedMa struggling due to its BN on VGG-9 [14].

In this experiment, the performance of FedMa is notably inferior to FedProx on the CIFAR-10 dataset, with a higher validation loss ($1.14 \pm 0.05$ at $C$ = 40%) and lower accuracy ($59.15 \pm 2.17$%) compared to FedProx ($0.92 \pm 0.06$ loss, $67.60 \pm 2.27$% accuracy). This contrasts with the results on MNIST and Fashion-MNIST, where FedMa exhibited performance parity or superiority over FedProx, as shown in Tables 3 and 4, and aligns with findings in [14]. The primary reason for this discrepancy lies in the model architecture used for CIFAR-10, specifically the VGG9 network, which incorporates BN layers while they were ignored in [14].

**Table 4. Mean Metrics on Fashion-MNIST across client fractions (Std with rolling window = 5).**

| Train Loss and Validation Loss | | | | | | |
|---|---|---|---|---|---|---|
| **Algorithm** | **Train Loss** | | | **Validation Loss** | | |
| | **10%** | **20%** | **30%** | **10%** | **20%** | **30%** |
| FedProx | 0.12 ± 0.03 | 0.10 ± 0.02 | 0.10 ± 0.01 | 0.53 ± 0.12 | 0.42 ± 0.06 | 0.38 ± 0.05 |
| FedMa | 0.12 ± 0.03 | 0.10 ± 0.02 | 0.10 ± 0.01 | 0.53 ± 0.12 | 0.42 ± 0.06 | 0.38 ± 0.05 |
| FedNolowe | 0.12 ± 0.04 | 0.10 ± 0.02 | 0.10 ± 0.02 | 0.55 ± 0.13 | 0.43 ± 0.06 | 0.38 ± 0.05 |
| FedAvg | 0.12 ± 0.04 | 0.10 ± 0.02 | 0.10 ± 0.01 | 0.60 ± 0.13 | 0.45 ± 0.07 | 0.39 ± 0.06 |
| FedAsl | 0.13 ± 0.04 | 0.11 ± 0.03 | 0.11 ± 0.02 | 0.66 ± 0.15 | 0.49 ± 0.09 | 0.44 ± 0.07 |
| **F1-score and Accuracy** | | | | | | |
| **Algorithm** | **F1-score** | | | **Accuracy (%)** | | |
| | **10%** | **20%** | **30%** | **10%** | **20%** | **30%** |
| FedProx | 0.80 ± 0.04 | 0.84 ± 0.02 | 0.86 ± 0.02 | 81.52 ± 3.33 | 84.96 ± 1.87 | 86.63 ± 1.53 |
| FedMa | 0.80 ± 0.04 | 0.84 ± 0.02 | 0.86 ± 0.02 | 81.49 ± 3.31 | 84.96 ± 1.90 | 86.63 ± 1.55 |
| FedNolowe | 0.79 ± 0.04 | 0.84 ± 0.02 | 0.86 ± 0.02 | 80.70 ± 3.64 | 84.73 ± 1.94 | 86.50 ± 1.57 |
| FedAvg | 0.78 ± 0.04 | 0.83 ± 0.03 | 0.85 ± 0.02 | 79.49 ± 3.63 | 84.36 ± 2.31 | 86.09 ± 1.72 |
| FedAsl | 0.76 ± 0.05 | 0.81 ± 0.04 | 0.83 ± 0.03 | 77.77 ± 3.91 | 82.58 ± 2.82 | 84.54 ± 2.04 |

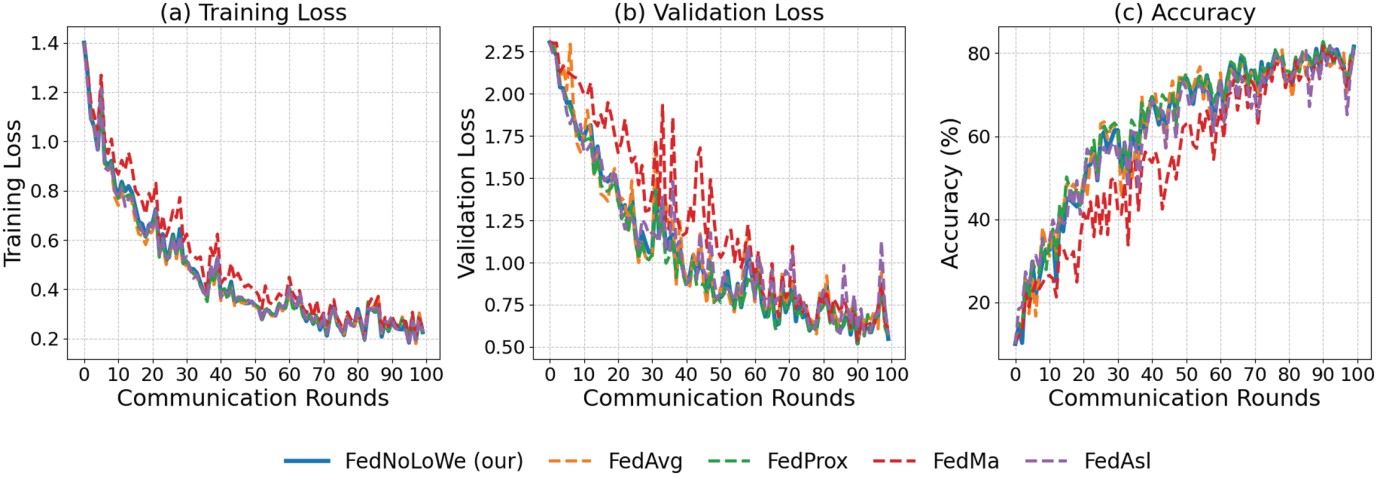

**Fig 9. Performance on CIFAR-10 with** $C = 20\%$ **clients per round: (a) training loss, (b) validation loss, (c) accuracy.**

## Computational efficiency

We evaluate the computational efficiency of FedNolowe against FedAvg [1], FedProx [13], FedMa [14], FedAsl [15] by measuring total floating-point operations (FLOPs [43]) per communication round, using PyTorch's profiler [42]. This metric accounts for both local training (forward and backward passes over $E$ epochs) and server-side aggregation operations specific to each method. Table 6 presents the total FLOPs for each model and client fraction, demonstrating FedNolowe's efficiency across diverse architectures and participation levels.

In Table 6, FedNolowe exhibits a robust computational efficiency, matching the FLOPs of FedAvg and FedAsl while significantly outperforming FedProx and FedMa. Specifically, FedNolowe reduces FLOPs by 17.55% to 18.95% compared to FedProx and by 21.26% to 40.10% compared to FedMa across all tested models and client fractions. For example, with LeNet-5 at a 30% client fraction, FedNolowe maintains 34.3 million FLOPs, whereas FedProx requires 41.6 million and FedMa demands 48.4 million. Similarly, for VGG-9 with 40% client

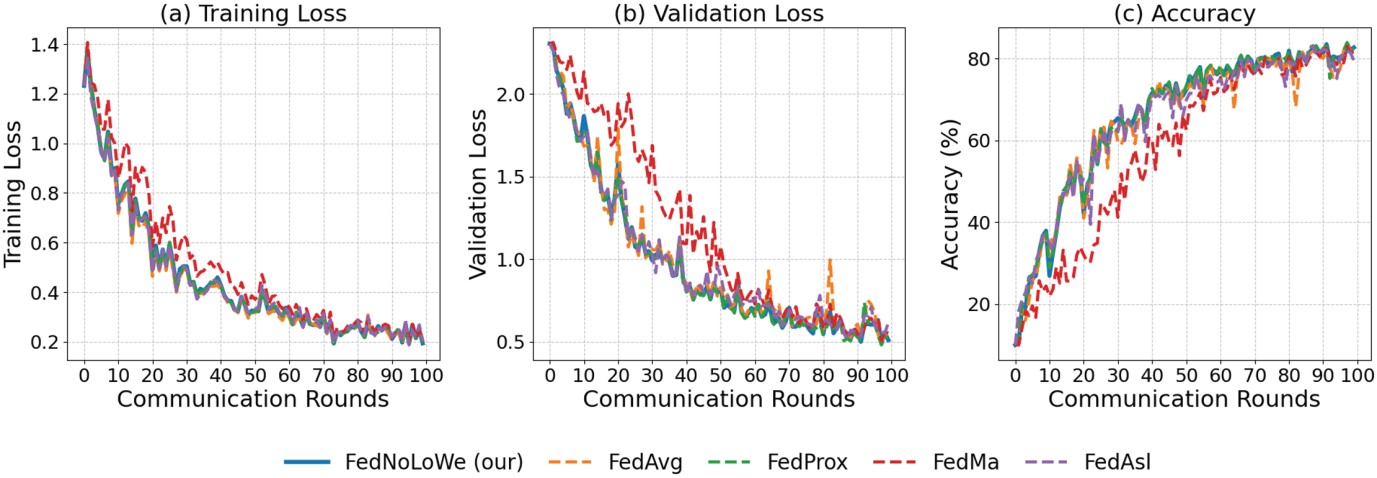

**Fig 10. Performance on CIFAR-10 with** $C$ **= 30% clients per round: (a) training loss, (b) validation loss, (c) accuracy.**

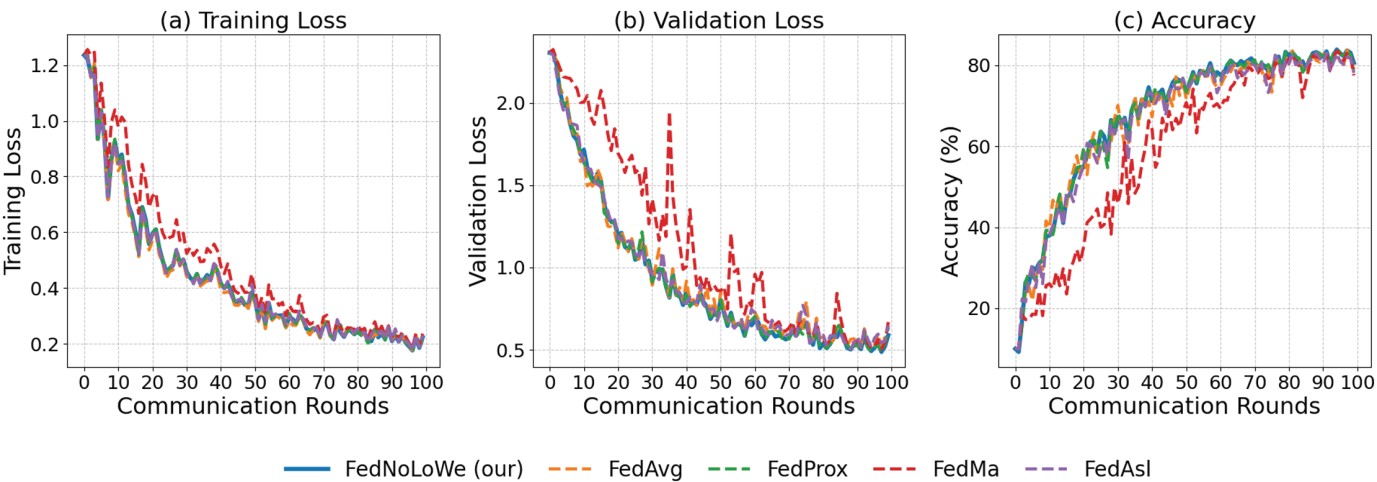

**Fig 11. Performance on CIFAR-10 with** $C$ **= 40% clients per round: (a) training loss, (b) validation loss, (c) accuracy.**

participation, FedNolowe utilizes 733.0 million FLOPs, compared to 834.0 million for Fed-Prox and 931.0 million for FedMa. These efficiency gains are driven by FedNolowe's streamlined approach, which avoids the resource-intensive proximal terms of FedProx and the complex neuron-matching process of FedMa, aligning with the needs of resource-constrained FL environments.

## Conclusion

In this paper, we introduce FedNolowe, a new aggregation method for FL that processes non-i.i.d data by dynamically adjusting the contribution of clients using two-step loss normalization. This method supports the global model stability by giving greater weight to high-performing clients and provides theoretical proof for convergence to a stationary point. Experiments on MNIST, Fashion-MNIST, and CIFAR-10 demonstrate its effectiveness,

**Table 5. Mean Metrics on CIFAR-10 Across Client Fractions (Std with rolling window = 5).**

**Train Loss and Validation Loss**

| Algorithm | Train Loss | | | Validation Loss | | |
|---|---|---|---|---|---|---|
| | 20% | 30% | 40% | 20% | 30% | 40% |
| FedNolowe | 0.46 ± 0.05 | 0.45 ± 0.04 | 0.43 ± 0.04 | 1.06 ± 0.09 | 0.97 ± 0.07 | 0.92 ± 0.05 |
| FedProx | 0.46 ± 0.05 | 0.45 ± 0.04 | 0.43 ± 0.04 | 1.03 ± 0.09 | 0.96 ± 0.06 | 0.92 ± 0.06 |
| FedAvg | 0.45 ± 0.05 | 0.44 ± 0.04 | 0.42 ± 0.04 | 1.08 ± 0.12 | 0.99 ± 0.09 | 0.94 ± 0.07 |
| FedAsl | 0.46 ± 0.05 | 0.45 ± 0.04 | 0.43 ± 0.04 | 1.09 ± 0.11 | 0.98 ± 0.07 | 0.93 ± 0.05 |
| FedMa | 0.52 ± 0.05 | 0.51 ± 0.05 | 0.49 ± 0.04 | 1.25 ± 0.12 | 1.17 ± 0.08 | 1.14 ± 0.05 |

**F1-score and Accuracy**

| Algorithm | F1-score | | | Accuracy (%) | | |
|---|---|---|---|---|---|---|
| | 20% | 30% | 40% | 20% | 30% | 40% |
| FedNolowe | 0.59 ± 0.04 | 0.64 ± 0.03 | 0.66 ± 0.02 | 62.07 ± 3.60 | 65.75 ± 2.57 | 67.61 ± 2.17 |
| FedProx | 0.61 ± 0.04 | 0.64 ± 0.02 | 0.66 ± 0.02 | 63.39 ± 3.42 | 65.92 ± 2.46 | 67.60 ± 2.27 |
| FedAvg | 0.59 ± 0.05 | 0.63 ± 0.03 | 0.65 ± 0.03 | 61.77 ± 4.18 | 64.82 ± 3.27 | 66.71 ± 2.83 |
| FedAsl | 0.58 ± 0.05 | 0.63 ± 0.03 | 0.65 ± 0.03 | 61.21 ± 3.89 | 64.86 ± 2.89 | 66.49 ± 2.55 |
| FedMa | 0.50 ± 0.04 | 0.54 ± 0.03 | 0.55 ± 0.02 | 54.95 ± 4.12 | 58.01 ± 2.79 | 59.15 ± 2.17 |

**Table 6. Total FLOPs per round (in millions) across models and client fractions.**

| Model | LeNet-5 | | | CNN | | | VGG-9 | | |
|---|---|---|---|---|---|---|---|---|---|
| | 10% | 20% | 30% | 10% | 20% | 30% | 20% | 30% | 40% |
| FedAvg | 11.0 | 22.6 | 34.3 | 24.8 | 51.0 | 77.1 | 363.0 | 548.0 | 733.0 |
| **FedNolowe** | **11.0** | **22.6** | **34.3** | **24.8** | **51.0** | **77.1** | **363.0** | **548.0** | **733.0** |
| FedAsl | 11.0 | 22.6 | 34.3 | 24.8 | 51.0 | 77.1 | 363.0 | 548.0 | 733.0 |
| FedProx | 13.4 | 27.5 | 41.6 | 30.6 | 62.5 | 94.4 | 414.0 | 624.0 | 834.0 |
| FedMa | 15.7 | 32.0 | 48.4 | 41.4 | 84.2 | 127.0 | 462.0 | 696.0 | 931.0 |

achieving increases in efficiency of up to 40 percent in computational complexity when compared to the state-of-the-art methods in different models. Comprehensive sensitivity analysis confirms competitive performance in heterogeneous whilst still being useful in i.i.d heterogeneity cases. Because of its low dependency design that only uses training losses, it can be deployed in constrained environments, enabling further research in variance-aware optimizations and practical use cases.

The inquiry, however, does not exhaust the spectrum of possibilities in this domain. Numerous techniques for normalizing and inverting losses in weighted aggregation merit consideration. For instance, normalization strategies might encompass min-max scaling, mean centering, or alternative approaches, while inversion methods could include exponential transformations, logarithmic adjustments, rank-based reweighting, and beyond. Investigating these diverse normalization and inversion frameworks presents a compelling avenue for future research. Furthermore, integrating loss-based weighting with feedback mechanisms in FL, as suggested in the recent survey by Le et al. (2024) [44], offers a promising opportunity to mitigate communication overhead, warranting further exploration in subsequent studies.

## Author contributions

**Conceptualization:** Duy-Dong Le, Anh-Khoa Tran.

**Data curation:** Duy-Dong Le.

**Formal analysis:** Duy-Dong Le.

**Investigation:** Duy-Dong Le.

**Methodology:** Duy-Dong Le, Tuong-Nguyen Huynh, Anh-Khoa Tran.

**Project administration:** Duy-Dong Le.

**Resources:** Duy-Dong Le.

**Software:** Duy-Dong Le.

**Supervision:** Tuong-Nguyen Huynh, Anh-Khoa Tran, Minh-Son Dao, Pham The Bao.

**Validation:** Duy-Dong Le.

**Visualization:** Duy-Dong Le.

**Writing – original draft:** Duy-Dong Le.

**Writing – review & editing:** Duy-Dong Le, Tuong-Nguyen Huynh, Anh-Khoa Tran, Minh-Son Dao, Pham The Bao.

## Appendices

## Sensitivity analysis of FedNolowe and FedAsl

People commonly use Eq (5) (division-based) to normalize an array by its sum, a method employed in FedAsl [15,18]. This approach enhances the aggregation weight of clients with lower losses [15] or correlations [18]. While effective in i.i.d scenarios where normalized values remain stable, it becomes susceptible in non-i.i.d settings compared to the subtraction-based approach (Eq (3)) used in FedNolowe. Figs 12, 13, 14, and 15 compare both methods.

Figs 12, 13, 14, and 15 present validation loss comparisons across three methods: FedNolowe (blue), FedAsl with $1 - d_k$ weighting adapted from FedNolowe (orange), and the original FedAsl with $1/d_k$ weighting [15] (green), under varying degrees of data heterogeneity.

In extremely and highly heterogeneous scenarios ($\alpha = 0.05$ and $0.1$), the subtraction-based methods—especially FedNolowe—consistently achieve lower and more stable validation loss. In contrast, the division-based approach shows large fluctuations, particularly during the early and middle rounds (e.g., 0, 10, and 21–25), due to its sensitivity to skewed data distributions.

As the distribution becomes more balanced ($\alpha = 0.5$), all methods converge quickly, but subtraction-based variants still exhibit slightly smoother learning curves. In the nearly i.i.d. case ($\alpha = 100$), the performance differences become negligible; all methods perform similarly with rapid convergence and minimal variance.

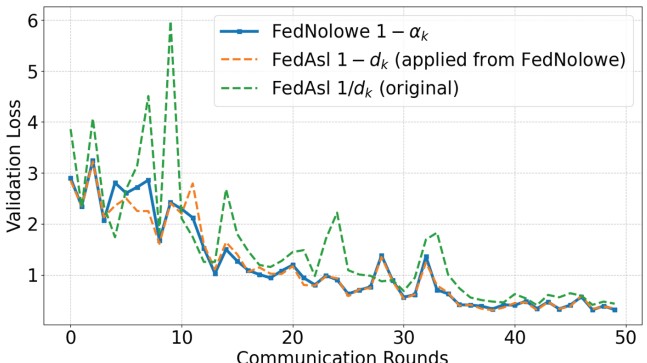

**Fig 12. Validation loss comparison with Dirichlet $\alpha = 0.05$ (extremely non-i.i.d.).**

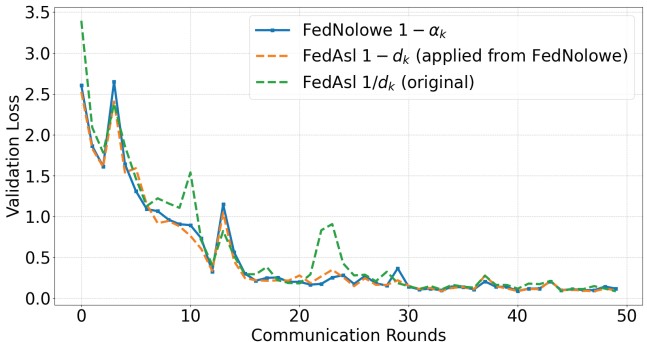

**Fig 13. Validation loss comparison with Dirichlet $\alpha = 0.1$ (highly non-i.i.d).**

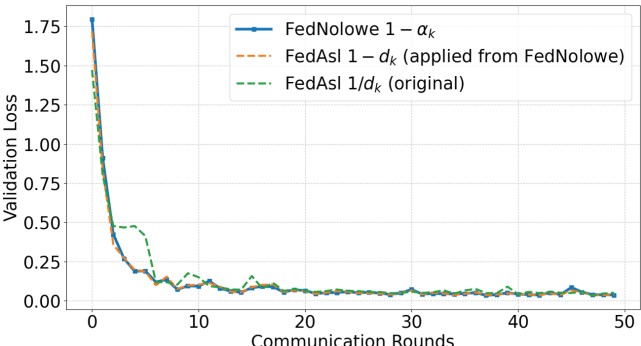

**Fig 14. Validation loss comparison with Dirichlet $\alpha = 0.5$ (moderately non-i.i.d).**

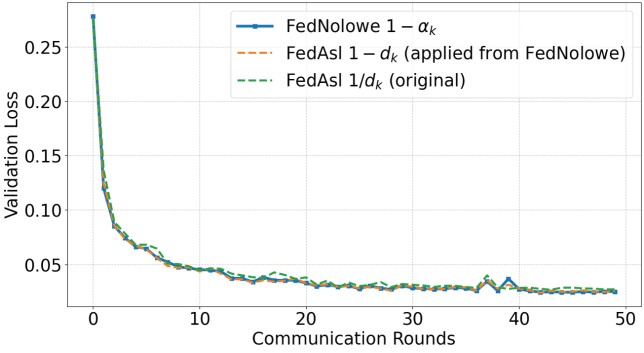

**Fig 15. Validation loss comparison with Dirichlet $\alpha = 100$ (nearly i.i.d).**

These trends highlight the robustness of the subtraction-based formulation in non-i.i.d. settings, while confirming that the division-based approach remains valid and effective in i.i.d. scenarios—consistent with the theoretical findings discussed later.

In the rest of this section, we provide a mathematical proof showing that our proposed weight assignment, defined in Eq (3)(subtraction-based utilized in FedNolowe), offerbased

approachater stability than the division-based of FedAsl's approach in Eq (5).

$$\alpha_k^t = \frac{1/\tilde{\mathcal{L}}_k^t}{\sum_{j\in S^t}(1/\tilde{\mathcal{L}}_j^t)} \quad \text{where} \quad \tilde{\mathcal{L}}_k^t = \frac{\mathcal{L}_k^t}{\sum_{j\in S^t}\mathcal{L}_j^t}. \tag{5}$$

## Step 1: Sensitivity analysis via partial derivatives

**For the subtraction-based (Eq (3)).** Let $f_k = 1 - \tilde{\mathcal{L}}_k^t$ and $g = \sum_{j\in S^t}(1 - \tilde{\mathcal{L}}_j^t)$, so $\alpha_k^t = \frac{f_k}{g}$. The sensitivity with respect to $\tilde{\mathcal{L}}_k^t$ is:

$$\frac{\partial \alpha_k^t}{\partial \tilde{\mathcal{L}}_k^t} = \frac{\frac{\partial f_k}{\partial \tilde{\mathcal{L}}_k^t}g - f_k\frac{\partial g}{\partial \tilde{\mathcal{L}}_k^t}}{g^2}, \tag{6}$$

where $\frac{\partial f_k}{\partial \tilde{\mathcal{L}}_k^t} = -1$ and $\frac{\partial g}{\partial \tilde{\mathcal{L}}_k^t} = -1$ (since $g$ depends on $\tilde{\mathcal{L}}_k^t$ through one term).

$$\frac{\partial \alpha_k^t}{\partial \tilde{\mathcal{L}}_k^t} = \frac{(-1)g - (1 - \tilde{\mathcal{L}}_k^t)(-1)}{g^2} = \frac{-\sum_{j\neq k}(1 - \tilde{\mathcal{L}}_j^t)}{g^2}. \tag{7}$$

The magnitude is bounded: $\left|\frac{\partial \alpha_k^t}{\partial \tilde{\mathcal{L}}_k^t}\right| \leq \frac{|S^t|-1}{g^2}$, since $0 < 1 - \tilde{\mathcal{L}}_j^t < 1$ and $g > 0$.

**For the division-base Eq (5).** The weight is:

$$\alpha_k^t = \frac{1/\tilde{\mathcal{L}}_k^t}{\sum_{j\in S^t}(1/\tilde{\mathcal{L}}_j^t)}.$$

Let $f_k = \frac{1}{\tilde{\mathcal{L}}_k^t}$ and $g = \sum_{j\in S^t}\frac{1}{\tilde{\mathcal{L}}_j^t}$, so $\alpha_k^t = \frac{f_k}{g}$. The sensitivity is:

$$\frac{\partial \alpha_k^t}{\partial \tilde{\mathcal{L}}_k^t} = \frac{\frac{\partial f_k}{\partial \tilde{\mathcal{L}}_k^t}g - f_k\frac{\partial g}{\partial \tilde{\mathcal{L}}_k^t}}{g^2}, \tag{8}$$

where $\frac{\partial f_k}{\partial \tilde{\mathcal{L}}_k^t} = -\frac{1}{(\tilde{\mathcal{L}}_k^t)^2}$ and $\frac{\partial g}{\partial \tilde{\mathcal{L}}_k^t} = -\frac{1}{(\tilde{\mathcal{L}}_k^t)^2}$.

$$\frac{\partial \alpha_k^t}{\partial \tilde{\mathcal{L}}_k^t} = \frac{\left(-\frac{1}{(\tilde{\mathcal{L}}_k^t)^2}\right)g - \left(\frac{1}{\tilde{\mathcal{L}}_k^t}\right)\left(-\frac{1}{(\tilde{\mathcal{L}}_k^t)^2}\right)}{g^2} = \frac{-\sum_{j\neq k}(1/\tilde{\mathcal{L}}_j^t)}{(\tilde{\mathcal{L}}_k^t)^2 g^2}. \tag{9}$$

The magnitude diverges as $\tilde{\mathcal{L}}_k^t \to 0^+$: $\left|\frac{\partial \alpha_k^t}{\partial \tilde{\mathcal{L}}_k^t}\right| \to \infty$.

## Step 2: Boundary behavior.

**Small loss ($\tilde{\mathcal{L}}_k^t \to 0^+$).** Subtraction-based approach $\alpha_k^t \to \frac{1}{\sum_{j\in S^t}(1-\tilde{\mathcal{L}}_j^t)}$, finite and bounded. Division-based approach $\alpha_k^t \to 1$ (if other losses are non-zero), indicating instability because there is no room for other clients.

**Large loss** ($\tilde{\mathcal{L}}_k^t \to 1$). Subtraction-based approach $\alpha_k^t \to 0$, robustly eliminating outliers. Division-based approach $\alpha_k^t \to \frac{1}{\sum_{j \in S^t}(1/\tilde{\mathcal{L}}_j^t)} > 0$, retaining influence.

## Step 3: Variance analysis

Subtraction-based approach variance of $\alpha_k^t$ is moderate, proportional to $1 - \tilde{\mathcal{L}}_k^t$. Division-based approach, Variance can be high, amplified by $1/\tilde{\mathcal{L}}_k^t$, especially near zero.

## Step 4: Stability in non-i.i.d vs. i.i.d contexts

The stability of the weight assignments depends on the data distribution across clients.

**Non-i.i.d stability.** In non-i.i.d settings, $\mathcal{L}_k^t$ varies significantly due to heterogeneous data. The division-based approach's sensitivity to small $\tilde{\mathcal{L}}_k^t$ leads to instability. Mathematically, as $\tilde{\mathcal{L}}_k^t \to 0^+$, the denominator $\sum_{j \in S^t}(1/\tilde{\mathcal{L}}_j^t)$ is dominated by $1/\tilde{\mathcal{L}}_k^t$, causing $\alpha_k^t \to 1$, which skews the aggregation disproportionately.

**I.i.d effectiveness.** In i.i.d settings, $\mathcal{L}_k^t$ are similar, reducing the variance of $\tilde{\mathcal{L}}_k^t$. The division-based approach's sensitivity is mitigated, as $\tilde{\mathcal{L}}_k^t$ values are close, preventing extreme weight imbalances. The Lipschitz continuity of $\alpha_k^t$ with respect to $\tilde{\mathcal{L}}_k^t$ holds better, ensuring stable aggregation.

## Remark

The subtraction-based approach is more stable in non-i.i.d due to its bounded sensitivity and robustness to loss variations. While unstable in non-i.i.d due to sensitivity to low losses, the division-based approach is effective in i.i.d where losses are consistent and outliers are rare.

## Detail convergence analysis of FedNolowe

### Empirical validation of gradient alignment

To support Assumption 4, we empirically evaluate the alignment between the aggregated weighted client gradients and the global gradient on the MNIST, Fashion-MNIST, and CIFAR-10 datasets.

At each communication round $t$, we compute the cosine similarity between the aggregated gradient $\sum_{k \in S^t} \alpha_k^t \nabla \mathcal{L}_k(w^t)$ and the full-batch global gradient $\nabla \mathcal{L}(w^t)$ as:

$$\cos(\theta^t) = \frac{\left\langle \nabla \mathcal{L}(w^t), \sum_{k \in S^t} \alpha_k^t \nabla \mathcal{L}_k(w^t) \right\rangle}{\|\nabla \mathcal{L}(w^t)\| \cdot \left\| \sum_{k \in S^t} \alpha_k^t \nabla \mathcal{L}_k(w^t) \right\|}.$$

We report the average and standard deviation of $\cos(\theta^t)$ across 50 rounds with C = 5 clients per round, each client performing 2 local epochs for MNIST, Fashion-MNIST, and CIFA-10 (Dirichlet concentration parameter $\alpha = 0.1$) as follows:

- **MNIST:** $0.3331 \pm 0.0596$
- **Fashion-MNIST:** $0.3421 \pm 0.0516$
- **CIFAR-10:** $0.3817 \pm 0.0457$

The cosine similarity remains consistently positive across all datasets. This empirical evidence supports the validity of Assumption 4, indicating that FedNolowe maintains strong gradient alignment during training, despite non-i.i.d client data.

## Proofs

Here, using Assumptions 1–4, we proof of Fednolowe's convergence as follow: Each client performs one stochastic gradient descent (SGD) step starting from the global model: $w_k^{t+1} = w^t - \eta g_k^t$, where $g_k^t$ is the stochastic gradient of $\mathcal{L}_k(w^t)$. The global update is:

$$w^{t+1} = \sum_{k \in S^t} \alpha_k^t w_k^{t+1}. \tag{10}$$

Thus, the change is:

$$w^{t+1} - w^t = \sum_{k \in S^t} \alpha_k^t (w_k^{t+1} - w^t) = -\eta \sum_{k \in S^t} \alpha_k^t g_k^t. \tag{11}$$

Using Assumption 1 (*L*-smoothness) on the global loss:

$$\mathcal{L}(w^{t+1}) \le \mathcal{L}(w^t) + \langle \nabla \mathcal{L}(w^t), w^{t+1} - w^t \rangle + \frac{L}{2} \|w^{t+1} - w^t\|^2. \tag{12}$$

Substituting (11) into (12):

$$\mathcal{L}(w^{t+1}) \le \mathcal{L}(w^t) - \eta \left\langle \nabla \mathcal{L}(w^t), \sum_{k \in S^t} \alpha_k^t g_k^t \right\rangle + \frac{L\eta^2}{2} \left\| \sum_{k \in S^t} \alpha_k^t g_k^t \right\|^2. \tag{13}$$

Taking expectations:

$$\mathbb{E}[\mathcal{L}(w^{t+1})] \le \mathcal{L}(w^t) - \eta \mathbb{E}\left[ \left\langle \nabla \mathcal{L}(w^t), \sum_{k \in S^t} \alpha_k^t g_k^t \right\rangle \right] + \frac{L\eta^2}{2} \mathbb{E}\left[ \left\| \sum_{k \in S^t} \alpha_k^t g_k^t \right\|^2 \right]. \tag{14}$$

**Gradient Term**: Since $\mathbb{E}[g_k^t] = \nabla \mathcal{L}_k(w^t)$, and using Assumption 4:

$$-\eta \mathbb{E}\left[ \left\langle \nabla \mathcal{L}(w^t), \sum_{k \in S^t} \alpha_k^t g_k^t \right\rangle \right] = -\eta \left\langle \nabla \mathcal{L}(w^t), \mathbb{E}\left[ \sum_{k \in S^t} \alpha_k^t g_k^t \right] \right\rangle \le -\eta \beta \|\nabla \mathcal{L}(w^t)\|^2, \tag{15}$$

noting that $\alpha_k^t$ depends on the loss, which is stochastic, but we assume the expectation aligns with the global gradient as per Assumption 4.

**Variance Term**: Expand the expectation:

$$\mathbb{E}\left[ \left\| \sum_{k \in S^t} \alpha_k^t g_k^t \right\|^2 \right] = \left\| \mathbb{E}\left[ \sum_{k \in S^t} \alpha_k^t g_k^t \right] \right\|^2 + \mathbb{E}\left[ \left\| \sum_{k \in S^t} \alpha_k^t (g_k^t - \nabla \mathcal{L}_k(w^t)) \right\|^2 \right]$$

$$\le \left\| \sum_{k \in S^t} \alpha_k^t \nabla \mathcal{L}_k(w^t) \right\|^2 + \sum_{k \in S^t} (\alpha_k^t)^2 \mathbb{E}[\|g_k^t - \nabla \mathcal{L}_k(w^t)\|^2]$$

$$\le G^2 + \sigma^2 \sum_{k \in S^t} (\alpha_k^t)^2, \tag{16}$$

where we bound $\left\| \sum_{k \in S^t} \alpha_k^t \nabla \mathcal{L}_k(w^t) \right\|^2 \leq G^2$ using Assumption 2, and $\mathbb{E}[\|g_k^t - \nabla \mathcal{L}_k(w^t)\|^2] \leq \sigma^2$ from Assumption 3. Since $\sum_{k \in S^t} \alpha_k^t = 1$, by Cauchy-Schwarz, $\sum_{k \in S^t}(\alpha_k^t)^2 \leq 1$, so:

$$\mathbb{E}\left[\left\|\sum_{k \in S^t} \alpha_k^t g_k^t\right\|^2\right] \leq G^2 + \sigma^2. \tag{17}$$

Combining terms:

$$\mathbb{E}[\mathcal{L}(w^{t+1})] \leq \mathcal{L}(w^t) - \eta\beta\|\nabla\mathcal{L}(w^t)\|^2 + \frac{L\eta^2}{2}(G^2 + \sigma^2). \tag{18}$$

Summing over $T$ rounds:

$$\sum_{t=0}^{T-1} \mathbb{E}[\mathcal{L}(w^{t+1})] - \mathbb{E}[\mathcal{L}(w^t)] \leq -\eta\beta \sum_{t=0}^{T-1} \mathbb{E}[\|\nabla\mathcal{L}(w^t)\|^2] + \frac{L\eta^2 T}{2}(G^2 + \sigma^2). \tag{19}$$

Since $\sum_{t=0}^{T-1}(\mathbb{E}[\mathcal{L}(w^{t+1})] - \mathbb{E}[\mathcal{L}(w^t)]) = \mathbb{E}[\mathcal{L}(w^T)] - \mathcal{L}(w^0) \leq \mathcal{L}(w^0)$ (assuming $\mathcal{L}$ is bounded below by 0), rearrange:

$$\frac{1}{T}\sum_{t=0}^{T-1} \mathbb{E}[\|\nabla\mathcal{L}(w^t)\|^2] \leq \frac{\mathcal{L}(w^0)}{\eta\beta T} + \frac{L\eta(G^2 + \sigma^2)}{2\beta}. \tag{20}$$

Choosing $\eta = \frac{1}{\sqrt{T}}$, the right-hand side becomes:

$$\frac{\mathcal{L}(w^0)}{\beta\sqrt{T}} + \frac{L(G^2 + \sigma^2)}{2\beta\sqrt{T}} \to 0 \text{ as } T \to \infty, \tag{21}$$

proving $\lim_{T\to\infty} \frac{1}{T}\sum_{t=0}^{T-1} \mathbb{E}[\|\nabla\mathcal{L}(w^t)\|^2] \to 0$, hence $\lim_{T\to\infty} \mathbb{E}[\|\nabla\mathcal{L}(w^t)\|^2] \to 0$.

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
