## [Decision Letter · Decision Letter 0]

6 May 2025

PONE-D-25-14711FedNolowe: A Normalized Loss-Based Weighted Aggregation Strategy for Robust Federated Learning in Heterogeneous EnvironmentsPLOS ONE

Dear Dr. Le,

Thank you for submitting your manuscript to PLOS ONE. After careful consideration, we feel that it has merit but does not fully meet PLOS ONE’s publication criteria as it currently stands. Therefore, we invite you to submit a revised version of the manuscript that addresses the points raised during the review process.   Please submit your revised manuscript by Jun 20 2025 11:59PM. If you will need more time than this to complete your revisions, please reply to this message or contact the journal office at plosone@plos.org. Please include the following items when submitting your revised manuscript:

In addition, it is recomended to review some rlated work to problem (1) since it is a finite-sum minimization problem arising from machine learning. For example, [Generalized Asymmetric Forward–Backward–Adjoint Algorithms for Convex–Concave Saddle-Point Problem, https://link.springer.com/article/10.1007/s10915-025-02802-7] also presented a stochastic method with convergence ensured; [Federated Learning: A Survey on Enabling Technologies, Protocols, and Applications, https://ieeexplore.ieee.org/document/9153560]  provides a comprehensive study of Federated Learning with an emphasis on enabling software and hardware platforms, protocols, real-life applications and use-cases.

Kind regards,

Academic Editor

PLOS ONE

2. Please update your submission to use the PLOS LaTeX template. The template and more information on our requirements for LaTeX submissions can be found at http://journals.plos.org/plosone/s/latex

Reviewers' comments:

1. Is the manuscript technically sound, and do the data support the conclusions?

Reviewer #1: Partly

2. Has the statistical analysis been performed appropriately and rigorously? 

Reviewer #1: No

3. Have the authors made all data underlying the findings in their manuscript fully available?

Reviewer #1: Yes

4. Is the manuscript presented in an intelligible fashion and written in standard English?

Reviewer #1: Yes

5. Review Comments to the Author

Reviewer #1: 1. Methodological novelty should be strengthened, like clarifying how FedNolowe’s two-stage normalization differs fundamentally from prior loss-based weighting (e.g., FedAsl’s inversion vs. FedNolowe’s subtraction).

2. Assumption 4 lacks some empirical support, while the cosine similarity between the client gradient and the global gradient should be calculated to verify this assumption and strengthen the validation of theoretical claims.

2. Non-IID data partitioning parameters (Dirichlet α=0.1 for MNIST vs. α=0.2 for other datasets) lack justification, and sensitivity analysis across α values is missing.

3. FedNolowe’s performance parity with FedProx/FedMa on MNIST/Fashion-MNIST but superiority on CIFAR-10 is not rigorously explained (e.g., model architecture interactions).

4. Metrics (training/validation loss, accuracy) are reported as averages over rounds, but lack variance estimates (standard deviation, error bands) or statistical tests (t-test, ANOVA) to confirm significance.

5. Raw training/validation loss trajectories, per-client weights, and code for FedNolowe are not provided, hindering reproducibility.

6. Minor issues require attention such as inconsistent terminology ("Manhattan normalization" in Abstract and "L1 normalization" in Section 3.2), and format issue of references ([1] and [15]).

6. PLOS authors have the option to publish the peer review history of their article (what does this mean?). If published, this will include your full peer review and any attached files.

Reviewer #1: No

---

## [Author Response · Author response to Decision Letter 1]

14 May 2025

Original Manuscript ID: PONE-D-25-14711

Original Article Title: “A Normalized Loss-Based Weighted Aggregation Strategy for Robust Federated Learning in Heterogeneous Environments”

May 10th, 2025

Dear Editorial board and Reviewers,

We sincerely thank the Editorial Board and Reviewers for providing the results and additional articles for us. We highly appreciate the reviewers' comments and have carefully reviewed the articles sent by the editorial board, incorporating them into the references of our paper as follows:

Article 1: "Generalized Asymmetric Forward--Backward--Adjoint Algorithms for Convex--Concave Saddle-Point Problem" has been included as reference number [36], cited at line 188 to strengthen the method of convergence analysis of our work, FedNolowe.

Article 2: "Federated Learning: A Survey on Enabling Technologies, Protocols, and Applications" has been included as reference number [8], cited at line 10 to support the idea of the impact of heterogeneous data on FL.

We have also re-submitted all the required files as requested:

- Response to Reviewers

- Revised Manuscript with Track Changes

- Revised Manuscript

Below are our responses to the revisions requested by the reviewer:

Reviewer #1:

1. Methodological novelty should be strengthened, like clarifying how FedNolowe’s two-stage normalization differs fundamentally from prior loss-based weighting (e.g., FedAsl’s inversion vs. FedNolowe’s subtraction).

We sincerely thank the reviewer for this helpful comment. To clarify, we have revised the Methodology section (lines 153–158) to highlight that FedNolowe adopts a two-stage normalization strategy—first scaling client losses across the round, then applying a subtraction-based inversion to compute weights. This contrasts with FedAsl’s direct division-based approach and helps avoid instability when losses are near zero or highly skewed in non-i.i.d. settings, resulting in more stable and bounded weights.

2. Assumption 4 lacks some empirical support, while the cosine similarity between the client gradient and the global gradient should be calculated to verify this assumption and strengthen the validation of theoretical claims.

We thank the reviewer for the thoughtful suggestion. To support Assumption 4, we conducted experiments measuring the cosine similarity between the aggregated weighted client gradients and the global gradient on MNIST, Fashion-MNIST, and CIFAR-10. As reported in Appendix Detail Convergence Analysis of FedNolowe (lines 638–651), the average cosine similarities across 50 communication rounds are as follows: MNIST (0.3331 ± 0.0596), Fashion-MNIST (0.3421 ± 0.0516), and CIFAR-10 (0.3817 ± 0.0457). These consistently positive values indicate strong alignment, thereby empirically supporting Assumption 4 even in non-i.i.d. settings.

3. Non-IID data partitioning parameters (Dirichlet α=0.1 for MNIST vs. α=0.2 for other datasets) lack justification, and sensitivity analysis across α values is missing.

- We thank the reviewer for the helpful comment. We have revised lines 220–222 to clarify our choice of Dirichlet concentration parameters for simulating non-i.i.d. settings. While FedMA and FedProx both adopted a relatively mild heterogeneity level with 𝛼=0.5, we intentionally used 𝛼=0.1 for MNIST to create a more challenging, highly heterogeneous setting, and 𝛼=0.2 for Fashion-MNIST and CIFAR-10 to reflect moderate heterogeneity while accounting for their greater data complexity. This results in skewed class distributions and variable sample sizes across clients, as illustrated in Figures 1 and 2, better simulating practical federated environments as suggested by Hsu et al. (2019).

- We thank the reviewer for pointing out the lack of sensitivity analysis across different Dirichlet 𝛼 values. In response, we have added Figures 12–15 in Appendix Sensitivity Analysis of FedNolowe and FedAsl, covering 𝛼=0.05, 0.1, 0.5, and 100, respectively. We also revised lines 579-593 to provide a more detailed explanation of how varying data heterogeneity levels affect the subtraction-based (FedNolowe) and division-based (FedAsl) approaches. The new results show that under highly heterogeneous conditions (𝛼=0.05 and 0.1), subtraction-based methods—especially FedNolowe—consistently achieve lower and more stable validation loss. Division-based weighting exhibits large fluctuations during early and mid-training rounds, highlighting its sensitivity to data skewness. As 𝛼 increases, performance across methods becomes similar; at 𝛼=100, where the data is nearly i.i.d., all methods converge rapidly with negligible differences. These findings confirm the robustness of the subtraction-based design in non-i.i.d. settings while reaffirming the validity of division-based weighting in i.i.d. cases, aligning well with our theoretical insights.

4. FedNolowe’s performance parity with FedProx/FedMa on MNIST/Fashion-MNIST but superiority on CIFAR-10 is not rigorously explained (e.g., model architecture interactions).

We thank the reviewer for this important observation. In response, we have added a detailed explanation in the revised manuscript (lines 396 to 403) to clarify why FedMA performs poorly on the CIFAR-10 dataset. Specifically, at a client fraction of 40%, FedMA exhibits a higher validation loss (1.14 ± 0.05) and lower accuracy (59.15% ± 2.17%) compared to FedProx, which achieves a lower loss (0.92 ± 0.06) and higher accuracy (67.60% ± 2.27%).

This result contrasts with the outcomes on MNIST and Fashion-MNIST, where FedMA demonstrates comparable or superior performance to FedProx, as shown in Tables 3 and 4. The performance degradation on CIFAR-10 primarily stems from the model architecture used—specifically the VGG9 network—which includes Batch Normalization layers. These layers were not considered in the original FedMA design and implementation, as noted in the original FedMA paper (Wang et al., 2020).

This clarification highlights the important interaction between data heterogeneity and model architecture complexity and explains why FedMA may struggle in deeper models like VGG9, despite its strong performance on simpler datasets.

5. Metrics (training/validation loss, accuracy) are reported as averages over rounds, but lack variance estimates (standard deviation, error bands) or statistical tests (t-test, ANOVA) to confirm significance.

We thank the reviewer for these important sugesstions. To address this, we have added standard deviation estimates computed using a rolling window of size 5 for all reported metrics (training loss, validation loss, F1-score, and accuracy) across all tables in the Results section from line 360 to line 93. These values provide a local measure of variability and offer a more informative view of the stability of each algorithm over training rounds. Additionally, we have revised the corresponding result descriptions in the text to reflect these variance estimates and support our conclusions with clearer evidence.

6. Raw training/validation loss trajectories, per-client weights, and code for FedNolowe are not provided, hindering reproducibility.

- Code Availability: We have added a footnote (1) in the Abstract linking to the publicly accessible GitHub repository containing the complete code for FedNolowe, ensuring full transparency and reproducibility. https://github.com/dongld-2020/fednolowe

- Raw training/validation loss trajectories, per-client weights: We thank the reviewer for the suggestion regarding raw training/validation loss trajectories and per-client weights. In response, we have added logging functionalities in the released codebase to enable users to record and inspect these raw outputs during training. We also attached the logs file of the additional experiments in the Sensitive Analysis section as Suporting Information.

7. Minor issues require attention such as inconsistent terminology ("Manhattan normalization" in Abstract and "L1 normalization" in Section 3.2), and format issue of references ([1] and [15]).

We thank the reviewer for these helpful suggestions.

- To ensure clarity and consistency, we have updated all instances of “Manhattan normalization” to the standard term “L1 normalization” throughout the manuscript. These changes are reflected at lines 41, Table 1 (above line 114), and in the Abstract.

- Regarding the formatting issues in references [1] and [15], we have revised them to comply with the required citation style. The corrected references now appear as follows:

[1] McMahan, Brendan; Moore, Eider; Ramage, Daniel; Hampson, Seth; Aguera y Arcas, Blaise. Communication-efficient learning of deep networks from decentralized data. In: Proceedings of the 20th International Conference on Artificial Intelligence and Statistics (AISTATS), PMLR; 2017. pp. 1273–1282.

[15] Li, Zexi; Lin, Tao; Shang, Xinyi; Wu, Chao. Revisiting weighted aggregation in federated learning with neural networks. In: Proceedings of the 40th International Conference on Machine Learning (ICML), PMLR; 2023. pp. 19767–19788.

These updates can be found on lines 446 and 488 in the revised manuscript.

We would like to once again express our sincere thanks to the Editors and Reviewers for their thoughtful feedback and constructive suggestions. We have carefully addressed all comments and revised the manuscript accordingly to improve its clarity, technical depth, and overall presentation.

We hope the revised version meets your expectations, and we remain open to any further suggestions. Thank you for your time and consideration.

Sincerely,

Duy-Dong Le,

Nguyen Huynh-Tuong,

Anh-Khoa Tran,

Minh-Son Dao,

and Pham The Bao

---

## [Decision Letter · Decision Letter 1]

6 Jun 2025

FedNolowe: A Normalized Loss-Based Weighted Aggregation Strategy for Robust Federated Learning in Heterogeneous Environments

PONE-D-25-14711R1

Dear Dr. Le,

We’re pleased to inform you that your manuscript has been judged scientifically suitable for publication and will be formally accepted for publication once it meets all outstanding technical requirements.

Kind regards,

Academic Editor

PLOS ONE

---

## [Editor Report · Acceptance letter]

PONE-D-25-14711R1

PLOS ONE

Dear Dr. Le,

I'm pleased to inform you that your manuscript has been deemed suitable for publication in PLOS ONE. Congratulations! Your manuscript is now being handed over to our production team.

Kind regards,

on behalf of

Dr. Jianchao Bai

Academic Editor

PLOS ONE